# Epidermal growth factor receptor activation is essential for kidney fibrosis development

Shirong Cao [1,2,6], Yu Pan[1,2,3,6], Andrew S. Terker[1,2], Juan Pablo Arroyo Ornelas [1,2], Yinqiu Wang[1,2], Jiaqi Tang[1,2], Aolei Niu[1,2], Sarah Abu Kar[1,2], Mengdi Jiang[1,2], Wentian Luo[1,2], Xinyu Dong [1,2], Xiaofeng Fan[1,2], Suwan Wang[1,2], Matthew H. Wilson [1,2,4], Agnes Fogo[5], Ming-Zhi Zhang [1,2] ✉ & Raymond C. Harris [1,2,4] ✉

Fibrosis is the progressive accumulation of excess extracellular matrix and can cause organ failure. Fibrosis can affect nearly every organ including kidney and there is no specific treatment currently. Although Epidermal Growth Factor Receptor (EGFR) signaling pathway has been implicated in development of kidney fibrosis, underlying mechanisms by which EGFR itself mediates kidney fibrosis have not been elucidated. We find that EGFR expression increases in interstitial myofibroblasts in human and mouse fibrotic kidneys. Selective EGFR deletion in the fibroblast/pericyte population inhibits interstitial fibrosis in response to unilateral ureteral obstruction, ischemia or nephrotoxins. In vivo and in vitro studies and single-nucleus RNA sequencing analysis demonstrate that EGFR activation does not induce myofibroblast transformation but is necessary for the initial pericyte/fibroblast migration and proliferation prior to subsequent myofibroblast transformation by TGF-ß or other profibrotic factors. These findings may also provide insight into development of fibrosis in other organs and in other conditions.

Fibrosis is a common pathological feature of most chronic inflammatory diseases. It is defined as the progressive accumulation of excess extracellular matrix and can cause organ failure as a result of scarring, loss of parenchyma, and destruction of organ structure[1,2]. Fibrosis can affect nearly every organ, and numerous clinical conditions can lead to organ fibrosis. Currently, there is no specific treatment for fibrosis[3], so there is a need to understand the underlying mechanisms of the development of fibrosis in order to develop potential therapeutic strategies.

In the kidney, tubulointerstitial fibrosis can result from incomplete recovery from acute kidney injury (AKI), toxic injury or other inflammatory insults. Recent studies indicate that following acute kidney injury, PDGFRα/PDGFRß+ (pericytes) and PDGFRα+/PDGFRß+ (kidney fibroblasts) cells migrate and proliferate and ultimately

differentiate into myofibroblasts, which are the major source of increased extracellular matrix expression[4,5]. It has been suggested that initial increases in the interstitial fibroblast population are important for regeneration and recovery from acute kidney injury[6]. However, incomplete recovery of kidney epithelia from acute injury and/or continued inflammatory injury can lead to persistence of myofibroblasts and development of tubulointerstitial fibrosis[7]. Although it is generally accepted that TGF-ß is an important mediator of myofibroblast transformation, mechanisms mediating the immediate fibroblast migration and proliferation are less well understood. In this regard, TGF-ß has been reported either not to induce or even to inhibit fibroblast proliferation[1].

The epidermal growth factor receptor (EGFR) is the prototypical member of a family of membrane-associated intrinsic tyrosine kinase

[1]Division of Nephrology and Hypertension, Department of Medicine, Nashville, TN, USA. [2]Vanderbilt Center for Kidney Disease, Nashville, TN, USA. [3]Division of Nephrology, Shanghai Ninth People's Hospital, Shanghai Jiao Tong University School of Medicine, Shanghai, China. [4]Veterans Affairs, Nashville, TN, USA. [5]Department of Pathology, Microbiology and Immunology, Vanderbilt University Medical Center, Nashville, TN, USA. [6]These authors contributed equally: Shirong Cao, Yu Pan. ✉e-mail: ming-zhi.zhang@vumc.org; ray.harris@vumc.org

receptors, the ErbB family, which includes EGFR (ErbB1), ErbB2, ErbB3, and ErbB4. EGFR is activated by its ligands, including EGF, TGF-α, HB-EGF, amphiregulin, betacellulin, epigen and epiregulin as well as by non-ligand mediated processes[8]. EGFR activation initiates activation of its intrinsic kinase domain and subsequent phosphorylation of specific tyrosine residues within the cytoplasmic tail. These phosphorylated residues serve as docking sites for a variety of signaling molecules whose recruitment leads to the activation of intracellular pathways controlling cell proliferation, differentiation, and apoptosis[9,10]. EGFR is expressed in multiple cell types in the kidney including podocytes, glomerular endothelial cells, mesangial cells, proximal tubules, thick ascending limbs, distal convoluted tubules, collecting ducts and medullary interstitial cells[8].

Selective deletion of proximal tubule EGFR delayed kidney functional recovery from ischemic injury[11], but persistent EGFR activation resulted in development of tubulointerstitial fibrosis[12–15]. Dsk5 mice have a mutation within the EGFR kinase domain that stabilizes the activation loop, producing a gain-of-function allele that increases basal receptor kinase activity. Without other intervention, they develop spontaneous and progressive kidney fibrosis[14]. Furthermore, mice with selective deletion of amphiregulin in the proximal tubule also developed less kidney fibrosis after kidney injury[16], while selective proximal tubule overexpression of human HB-EGF led to development of spontaneous kidney fibrosis. This fibrosis was prevented by administration of an EGFR tyrosine kinase inhibitor or by crossing with *waved2* mice, which have deficient EGFR tyrosine kinase activity[17]. However, how EGFR activation mediated increased interstitial fibrosis was not addressed in these previous studies. Therefore, the goal of the present studies was to determine the potential underlying mechanisms by which EGFR activation can mediate fibroblast activation in conditions leading to progressive kidney interstitial fibrosis.

We now find that EGFR expression increases in interstitial myofibroblasts in human and mouse fibrotic kidneys. Selective EGFR deletion in the fibroblast/pericyte population inhibits interstitial fibrosis in response to kidney injury by unilateral ureteral obstruction, ischemia or nephrotoxins. EGFR activation does not induce myofibroblast transformation but is necessary for the initial pericyte/fibroblast migration and proliferation prior to subsequent myofibroblast transformation by TGF-ß or other profibrotic factors.

## Results

### Mice with iRhom2 deletion had less EGFR activation in myofibroblasts and developed less kidney fibrosis in response to kidney injury

iRhom2 is a member of the Rhomboid intramembrane protein family and is an essential mediator of ADAM17-mediated release of a number of active soluble growth factors and cytokines from their membrane-anchored forms, including the EGFR ligands HB-EGF and amphiregulin, in addition to TNF-α[18–25]. Unlike its paralog, iRhom1, which is ubiquitously expressed, iRhom2 has been previously reported to be largely confined to myeloid cells[26,27]. In this regard, previous studies have demonstrated a role for macrophage involvement in the development of kidney fibrosis in response to kidney injury[28,29]. We found that both total kidney and myeloid iRhom2 mRNA expression increased in response to unilateral ureteral obstruction (UUO) (Fig. 1a, b). iRhom1 mRNA also increased following UUO (Supplementary Fig. S1). Furthermore, in addition to iRhom2 expression in inflammatory macrophages, we unexpectedly found increased mRNA expression of iRhom2 in isolated kidney myofibroblasts after UUO (Fig. 1c). Immunofluorescent staining confirmed iRhom2 expression in α-SMA+ myofibroblasts (Fig. 1d).

We utilized iRhom2−/− mice to determine its role in development of kidney fibrosis after ischemic injury. Four weeks after ischemic injury, iRhom2−/− mouse kidneys had significantly lower mRNA levels of profibrotic and fibrotic components, including *Acta2*, *Col1a1*, *Col3a1*, *Fn*,

and *Tgfb1* (Fig. 1e). Immunoblotting and Picrosirius red staining and qPCR confirmed decreased kidney expression of α-SMA and collagen IV (Fig. 1f), decreased fibrosis (Fig. 1g), less macrophage infiltration and lower proinflammatory cytokines/chemokines compared to WT mice (Fig. 1h, i). This phenotype is similar to phenotypes previously reported in mice with global ADAM17 deletion[30]. We also saw less kidney injury, as indicated by lower NGAL and KIM-1 expression and less tubulointerstitial fibrosis 7 days after UUO in iRhom2−/− mice, as indicated by both Picrosirius red and Masson Trichrome stain (Supplementary Fig. S2a, b). Of note, using confocal microscopy, we found that although there were no differences in total EGFR expression in α-SMA⁺ myofibroblasts (Supplementary Fig. S2c) and EGFR activation in tubular epithelial cells (phospho-EGFR expression in plasma membrane) (Fig. 1j) in WT mice and iRhom2−/− mice 7 days after UUO, EGFR activation in myofibroblasts (phospho-EGFR colocalization with α-SMA) could be detected in WT mice but was minimal in iRhom2−/− mice (Fig. 1j).

### EGFR and its ligands were highly expressed in kidney myofibroblasts in murine unilateral ureteral obstruction

In control mouse kidneys, EGFR was primarily expressed in tubular epithelial cells, with minimal expression in α-SMA+ cells (Fig. 2a). Seven days after initiation of UUO, EGFR expression increased in kidney epithelial cells, but in addition, there was a significant increase in EGFR expression in interstitial α-SMA+ cells, indicating increased myofibroblast EGFR expression (Fig. 2b). Kidney transcripts were increased for *Egfr*, and EGFR ligands, including *Hbegf*, *Areg*, *Btc*, *Tgfa* and *Ereg*, as well as Adam17 (Fig. 2d–j), while expression levels of all these mRNA species were essentially unchanged after UUO in the contralateral kidney (Supplementary Fig. S3).

We further investigated the cell types expressing components of the EGFR signaling pathway. Immunofluorescence indicated increased HB-EGF expression in tubular epithelial cells and myofibroblasts (colocalization with α-SMA) 3 days and 7 days after UUO (Supplementary Fig. S4a). RNAscope indicated AREG mRNA in tubular epithelial cells and myofibroblasts (colocalization with α-SMA gene, Acta2) 7 days after UUO (Supplementary Fig. S4b). Of note ADAM17 was primarily expressed in myofibroblasts 3 days after UUO, and its expression was markedly lower in FibEGFR-/- mice than in WT mice (Supplementary Fig. S4c).

### Single-nucleus RNA sequencing (snRNAseq) analysis of fibroblast and myofibroblast subtypes following UUO

Since our results, as well as those of others indicated increased kidney expression of multiple EGFR ligands following acute injury[20–25], alteration of expression of any single EGFR ligand may not fully encompass the effect of pericyte/fibroblast EGFR activation. Therefore, we chose to investigate the potential role of altering pericyte/fibroblast EGFR expression in kidney fibrogenesis, utilizing mice with selective and inducible EGFR deletion in pericytes/fibroblasts/myofibroblasts. We generated PDGFRβ-Cre/ERT2; mCherry (WT) and PDGFRβ-Cre/ERT2; mCherry; EGFR^f/f (FibEGFR−/−) mice, treated both strains with tamoxifen, and studied their response to UUO (Fig. 3a). In the absence of injury, mCherry⁺ cells, which include PDGFRβ+ pericytes and PDGFRβ+ fibroblasts, predominantly surrounded blood vessels (Fig. 3b). Of Note, in the detailed photo, nuclei are pseudocolored as white in order to allow the localization in different cell types. We isolated kidney myofibroblasts at day 7 after UUO from both WT and FibEGFR−/− mice. *Egfr* mRNA levels were markedly lower in kidney myofibroblasts isolated from FibEGFR-/- mice compared to WT mice ($5.75 \pm 0.80$ vs. $37.08 \pm 3.98$ of WT, $n = 6$, $P < 0.001$), confirming effective deletion (Fig. 3c). Although mCherry⁺ cells in the interstitium of WT mice were sparse at baseline, seven days after UUO they were markedly increased, but FibEGFR−/− mice still had significantly fewer mCherry+ cells compared to WT mice (mCherry⁺ cells/field: $114.6 \pm 6.6$ vs. $39.8 \pm 3.7$, $n = 5$, $P < 0.001$) (Fig. 3d).

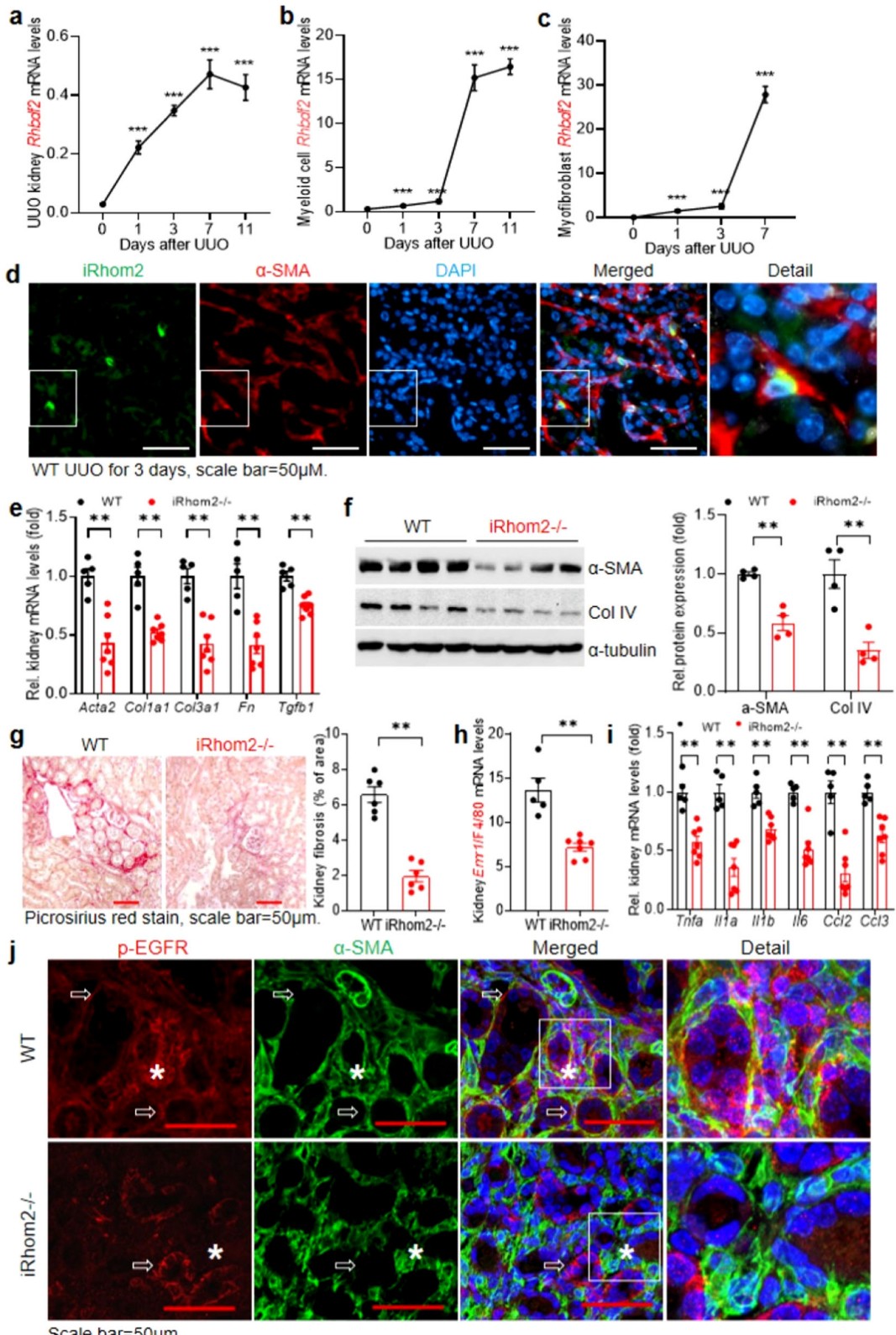

snRNAseq has clear advantages over single cell RNA sequencing (scRNAseq) to analyze transcriptomics of adult kidney because it allows interrogation of gene activity of activated fibroblasts and other rare cell types[31–34]. To delineate the effect of deletion of EGFR in the fibroblast population, we performed snRNAseq on kidney samples from WT and FibEGFR−/− mice one and three days after UUO, with each sample being a mixture of kidneys from 3–5

individual mice of the same group. Both cell counts and gene numbers were acceptable for further analysis in all samples (Supplementary Fig. S5). Canonical markers of kidney cell populations to identify the major cell types indicated that one day after UUO, the percentage of cells localized to each cell type was similar in the samples from wild type and FibEGFR−/− mice. In contrast, three days after UUO, the percentage of the fibroblast compartment in the total

**Fig. 1 | Kidney iRhom2 expression increased and contributed to the development of kidney fibrosis after kidney injury.** Mice underwent unilateral ureteral obstruction (UUO) or ischemic AKI. *Rhbdf2* (iRhom2) transcripts increased gradually in whole kidney (**a**), isolated kidney macrophages (**b**), and isolated myofibroblasts (**c**) at 7 days after UUO. *n* = 7 and 9. **d** Representative photomicrographs of 3 independent experiments showed that iRhom2 was colocalized with α-SMA in WT mice 3 days after UUO. Scale bar = 50 μm. **e** Kidney profibrotic and fibrotic genes, including *Acta2, Col1a1, Col3a1, Fn,* and *Tgfb1,* were lower in iRhom2−/− mice than WT mice 3 weeks after AKI. *n* = 5 and 7. **f** There was less immunoreactive α-SMA and Collagen IV (Col IV) expression in kidneys of iRhom2−/− mice 3 weeks after AKI.

*n* = 4. **g** Quantitative Picrosirius red staining showed less kidney fibrosis in iRhom2−/− mice 3 weeks after AKI. *n* = 6. Kidney mRNA levels of *Emr1* (F4/80) (**h**) and proinflammatory cytokines, including *Tnfa, Il1a, Il1b, Il6, Ccl2,* and *Ccl3* were lower in iRhom2-/- mice than WT mice 3 weeks after AKI. *n* = 5 and 7. **j** Confocal microscopy data showed that iRhom2−/− mouse kidney had minimal p-EGFR expression in myofibroblasts (asterisk) but similar p-EGFR expression in epithelial cells (arrows) compared to WT mice 7 days after UUO. (representative of 3 studies) Scale bar = 50 μM. Data are means ± SEM, **P < 0.01, ***P < 0.001, analyzed using 2-way ANOVA followed by Tukey's post hoc test for (**a**−**c**); two tailed Student's *t* test (**e**−**i**).

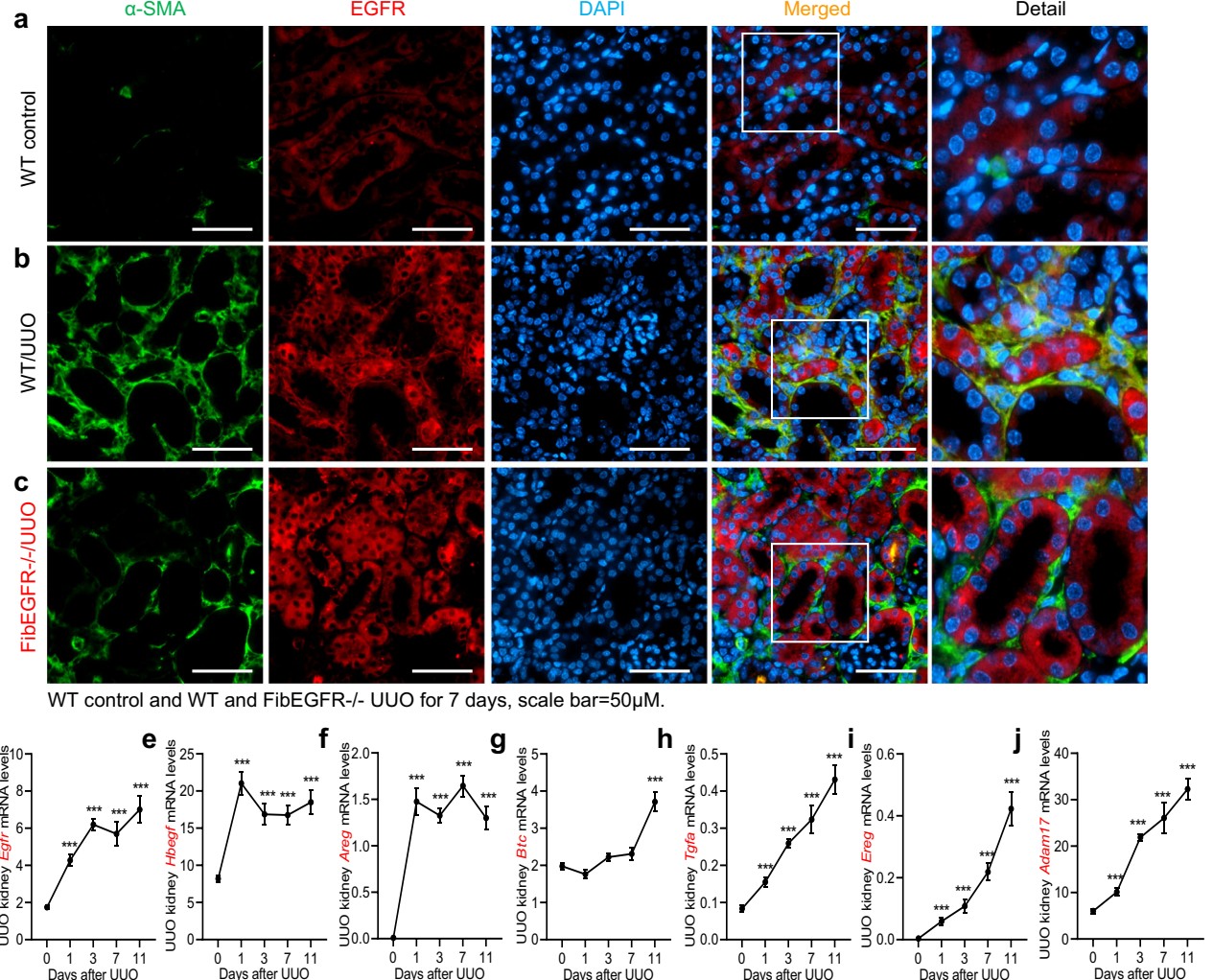

**Fig. 2 | Myofibroblast EGFR signaling pathway was activated following unilateral ureteral obstruction (UUO).** Representative photomicrographs of 3 independent experiments showed that EGFR+ myofibroblasts (colocalization with α-SMA, a marker of myofibroblasts) were increased in WT mice 7 days after UUO (**a, b**). Of note, both α-SMA+ myofibroblasts and EGFR+ myofibroblasts were

decreased in FibEGFR−/− mice compared to WT mice after UUO for 7 days (**c**). Scale bar = 50 μm. **d**−**j** Kidney mRNA levels of *Egfr, Hbegf, Areg, Btc, Tgfa, Ereg,* and *Adam17* were all progressively increased after UUO. *n* = 7 and 8. Data are means ± SEM, ***P < 0.001, analyzed using two-way ANOVA followed by Tukey's post hoc test for all.

cells increased to 13.9% in WT mice while in the FibEGFR−/− samples it was only 4.3% (Fig. 3e–g). A violin plot demonstrated that PDGFRß+ myofibroblasts (MF) were lower in FibEGFR−/− (KO) mice than WT mice three days after UUO (Fig. 3h). A volcano plot indicated higher expression levels of extracellular matrix- and myofibroblast-associated genes in WT mice compared to FibEGFR−/− mice on day 3 (Fig. 3i).

The tSNE plot identified 7 clusters of fibroblasts designated as Fibroblast 1 and 2 (Fib1-2) and Myofibroblast 1−5 (MF1-5) (Fig. 4a, b).

As indicated in the violin plot of WT, *Pdgfra* and *Pdgfrb* transcripts were expressed in all 7 clusters, although their relative expression was lower in the Fibroblast 1and 2 clusters (Fig. 4c). EGFR mRNA expression was evident in all fibroblast clusters (Fig. 4c). A violin plot of myofibroblast-associated genes indicated increased expression of *Col1, Dcn, Acta2, Tagln,* and *Creb5* in the myofibroblast clusters, with the consistently highest levels of expression in the Myofibroblast 5 cluster (Fig. 4d). On day 1 after UUO, 96−98% of cells in the fibroblast component were in Fibroblast 2 and Myofibroblast 2 clusters, and

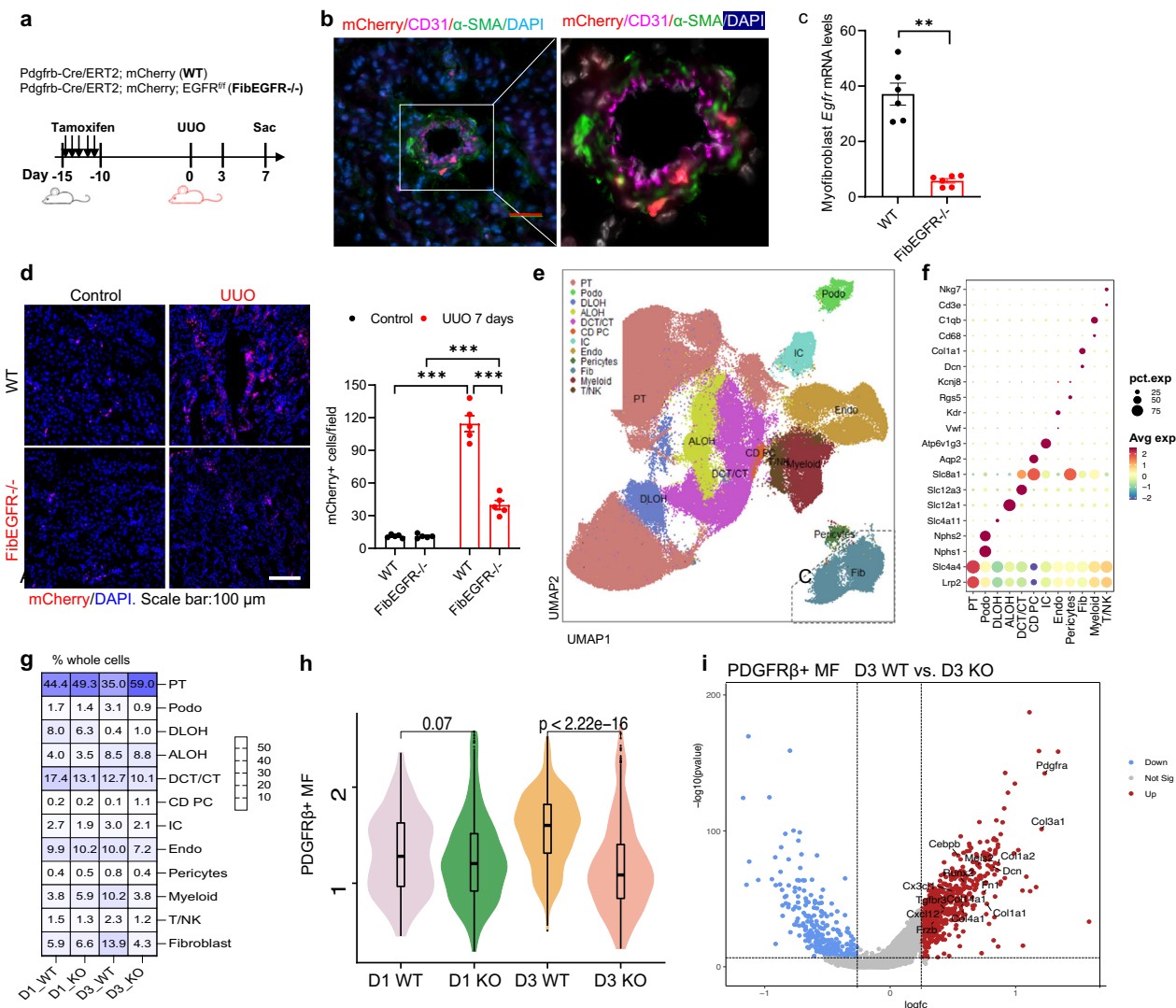

**Fig. 3 | Fibroblast EGFR deficiency led to decreased myofibroblasts following unilateral ureteral obstruction (UUO). a** Schematic of experimental protocol. **b** Under normal conditions, mCherry-positive cells (both pericytes and fibroblasts) were detected surrounding blood vessels. (representative of 3 studies) Scale bar = 50 μm. **c** Myofibroblasts isolated from FibEGFR−/− mice after UUO for 7 days expressed markedly lower mRNA levels of *Egfr*. *n* = 6. **d** Marked increases in mCherry+ cells seen in WT mice at day 7 after UUO were significantly attenuated in FibEGFR−/− mice. *n* = 5. Scale bar = 100 μm. **e** Single nucleus RNAseq (snRNAseq) analysis identified clusters of kidney cells. **f** Canonical markers of kidney cell populations were used to identify major cell types in the kidney: podocyte (*Nphs1*), endothelial cells (*vWF*), proximal tubule (PT) (*Lrp2, slc4a4*), descending limb of Henle's loop (DLOH) (*Slc4a11*), ascending limb of Henle's loop (ALOH) (Slc12a1),

distal convoluted tubules (DCT) (*Slc12a3*), collecting duct principal cell (CD-PC) (*Slc8a1,Aqp2*), collecting duct intercalated cell (CD-IC) (*Atp6v1g3*), pericytes (*Rgs5*), fibroblast (Fib) (*Dcn,Col1a1*), myeloid cell (*Cd68,C1qb*) and T/NK cell (*CD3e, Nkg7*). **g** Cells with expression of canonical genes for fibroblasts were significantly decreased in FibEGFR−/− mice at day 3 after UUO. **h** Violin plot indicated less PDGFRβ+ myofibroblasts (MF) in FibEGFR−/− mice (D3 KO) compared to WT (D3 WT) at day 3 after UUO. **i** A volcano plot indicated higher expression of extracellular matrix- and myofibroblast-associated genes (including *Pdgfra, Col1a1, Col1a2, Col3a1, Col4a1, Col14a1, Fn*) in WT mice than FibEGFR−/− mice at day 3 after UUO. Data are means ± SEM, **$P < 0.01$, ***$P < 0.001$, analyzed using two tailed Student's *t* test for (**c**); 2-way ANOVA followed by Bonferroni's post hoc test for (**d**).

there were comparable levels of cells in these two clusters in WT mice and FibEGFR−/− mice (Fig. 4e). In contrast, on day 3, the percentage of Fibroblast 2 and Myofibroblast 2 clusters in total cells decreased to very low levels in both WT mice and FibEGFR−/− mice. Of note, the percentage of Fibroblast 1 cluster in total cells was markedly increased in WT mice compared to FibEGFR−/− mice (9.10% vs. 0.66% of all cells). There were also greater increases in the Myofibroblast 3 and Myofibroblast 4 clusters in WT mice compared to FibEGFR−/− mice. Myofibroblast 5, the cluster of mature myofibroblasts, was low in both WT and FibEGFR−/− samples at 3 days, consistent with the relatively early nature of the injury. A stacked bar plot showed more cells from clusters Fib1, MF1, MF3, and MF4 in WT mice than FibEGFR−/− mice 3 days following UUO (Fig. 4f).

## Mice with fibroblast EGFR deletion developed less kidney fibrosis in UUO

Following UUO, FibEGFR−/− mice had significantly decreased kidney expression of EGFR protein compared to corresponding WT mice, as indicated by immunoblotting (Fig. 5a) as well as minimal EGFR immunofluorescence in α-SMA+ cells (Fig. 2c). Immunofluorescent phospho-EGFR was not present in α-SMA+ cells in WT kidneys at baseline and day 1 after UUO but was detected in α-SMA+ cells of WT at 3 days after UUO although not in FibEGFR−/− mice at any time point (Supplementary Fig. S6). Seven days after UUO, FibEGFR−/− mice developed less kidney fibrosis compared to WT, as indicated by qPCR and immunoblotting of profibrotic and fibrotic components, including α-SMA, PDGFRβ, Col I, Col IV, and FN (Fig. 5b, c). Myofibroblasts

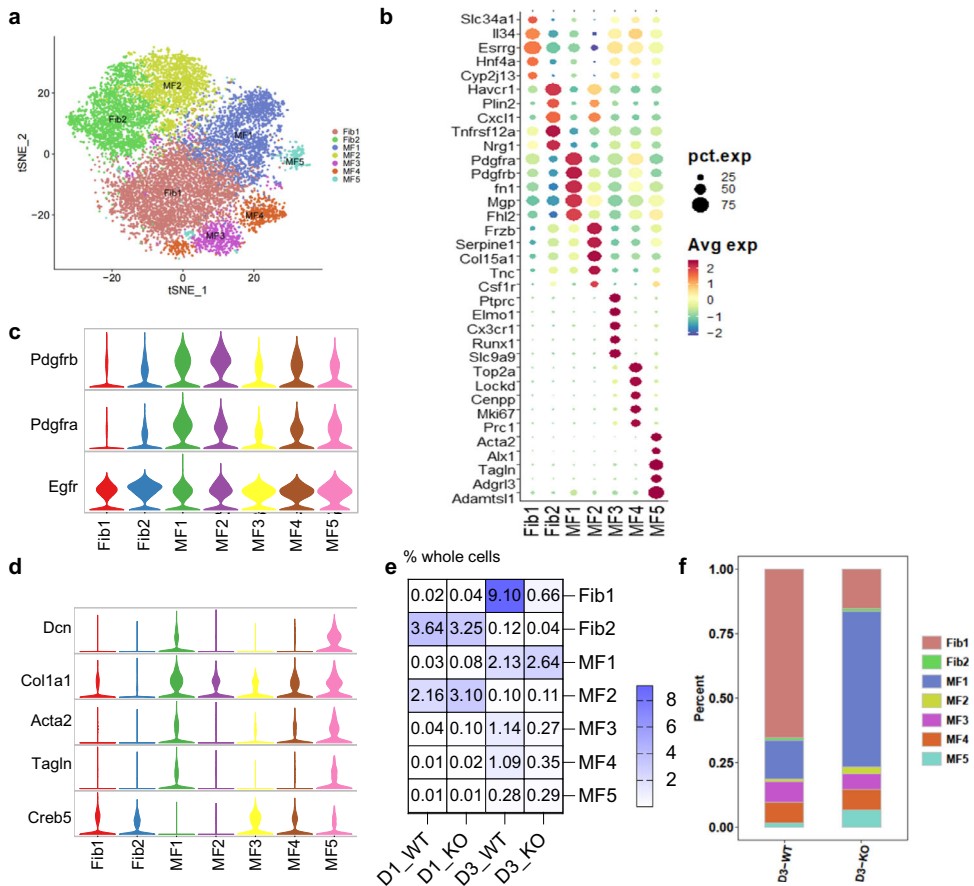

**Fig. 4 | Fibroblast EGFR deficiency led to decreased expansion of specific subclusters of fibroblasts and myofibroblasts following unilateral ureteral obstruction (UUO). a** The tSNE plot identifies 7 clusters of fibroblasts corresponding to fibroblasts (Fib) and myofibroblasts (MF): Fib 1, 2 and MF1 - 5. **b** Bubble plot indicated representative genes in each subcluster: *Il34, Hfn4a* (Fib1); *Havcr1, Cxcl1, Nrg1* (Fib2); *Pdgfra, Pdgfrb, Fn1* (MF1); *Frzb, Serpine1, Csf1r* (MF2); *Elmo1, Cx3cr1* (MF3); *Top2a, MKi67* (MF4); *Acta2, Tagln* (MF5). **c** Violin plot indicated that *Pdgfrb, Pdgfra* were mainly expressed in MFs while *Egfr* was expressed in each subcluster. **d** Violin plot indicated the expression of classical myofibroblast markers such as *Col1a1* and *Acta2.* **e** Compared to WT mice, FibEGFR-/- mice had significantly reduced Fib1, MF3 and MF4 subclusters at day 3 after UUO. **f** The stacked bar plots for both WT and FibEGFR-/- at day 3 after UUO.

isolated from FibEGFR−/− mouse kidney had lower *Col1a1* and *Col4a1* mRNA levels compared to WT mouse kidney after 7 days of UUO (Fig. 5d). Quantitative α-SMA immunostaining (Fig. 5e) and Picrosirius red staining (Fig. 5f) confirmed less kidney fibrosis in FibEGFR−/− mice.

As noted, immune cells play important roles in the development of kidney fibrosis; however, kidney macrophage and lymphocyte infiltration were comparable in WT and FibEGFR−/− mice 7 days after UUO, indicated by qPCR and quantitative immunostaining (Supplementary Fig. S7a, b). The mRNA levels of proinflammatory cytokines/chemokines in the kidney, including *Tnf, Il1a, Il1b, Ccl2, Ccl3, Il23a* and *Ifng* and the tubular injury marker *Kim-1* were also comparable in WT and FibEGFR−/− mice (Supplementary Fig. S7c, d), as wells as the mRNA levels of proinflammatory cytokines/chemokines in isolated kidney macrophages (Supplementary Fig. S7e).

### Mice with fibroblast EGFR deletion developed less kidney fibrosis after ischemic or toxin injury

In a mouse model of ischemia/reperfusion injury (IRI) (Fig. 6a), kidney functional recovery was comparable in WT and FibEGFR−/− mice after injury (Fig. 6b). On day 28 after injury, the mRNA levels of kidney injury markers (Kim-1 and Ngal) (Fig. 6c) and proinflammatory cytokines were also comparable in WT and FibEGFR−/− mice (Fig. 6d). Although functional recovery and tubule injury markers were not different, similar to what was seen in the UUO model, FibEGFR−/− mice developed less fibrosis compared to WT mice, as indicated by immunoblotting of α-SMA, Col I and FN (Fig. 6e),

quantitative α-SMA immunostaining (Fig. 6f) and Picrosirius red staining (Fig. 6g).

We also investigated whether FibEGFR−/− mice were also protected against the development of kidney fibrosis in response to toxin models of interstitial fibrosis. Mice were administered folic acid at a dose of 0.25 mg/g intraperitoneally and sacrificed after 19 days (Fig. 7a)[35]. There were similar initial peaks and declines of BUN after folic acid administration in FibEGFR−/− and WT mice (Fig. 7b). The mRNA levels of kidney injury marker, macrophages, and proinflammatory cytokines in both kidney and isolated kidney macrophages were also similar in FibEGFR−/− and WT mice (Fig. 7c–g). Kidney myofibroblasts isolated from FibEGFR−/− mice had lower levels of *Egfr, Acta2, Col1a1* and *Col4a1* transcripts (Fig. 7h), and FibEGFR−/− mice had less kidney fibrosis compared to WT mice, as indicated by immunoblotting of α-SMA, Col I and FN (Fig. 7i), quantitative α-SMA immunostaining (Fig. 7j) and Picrosirius red staining (Fig. 7k).

In a second model of toxic nephropathy, adenine was administered in the diet, and mice were sacrificed at the indicated time points (Supplementary Fig. S8a). Analysis of kidney fibroblasts/myofibroblasts at day 3 and day 14 after adenine treatment again indicated that FibEGFR-/- mouse kidney fibroblasts/myofibroblasts had effective EGFR deletion and lower levels of *Acta2, Col1a1, Col4a1,* and *Fn* transcripts (Supplementary Fig. S8b–f), and quantitative α-SMA immunostaining and Picrosirius red staining indicated less interstitial fibrosis (Supplementary Fig. S8g, h).

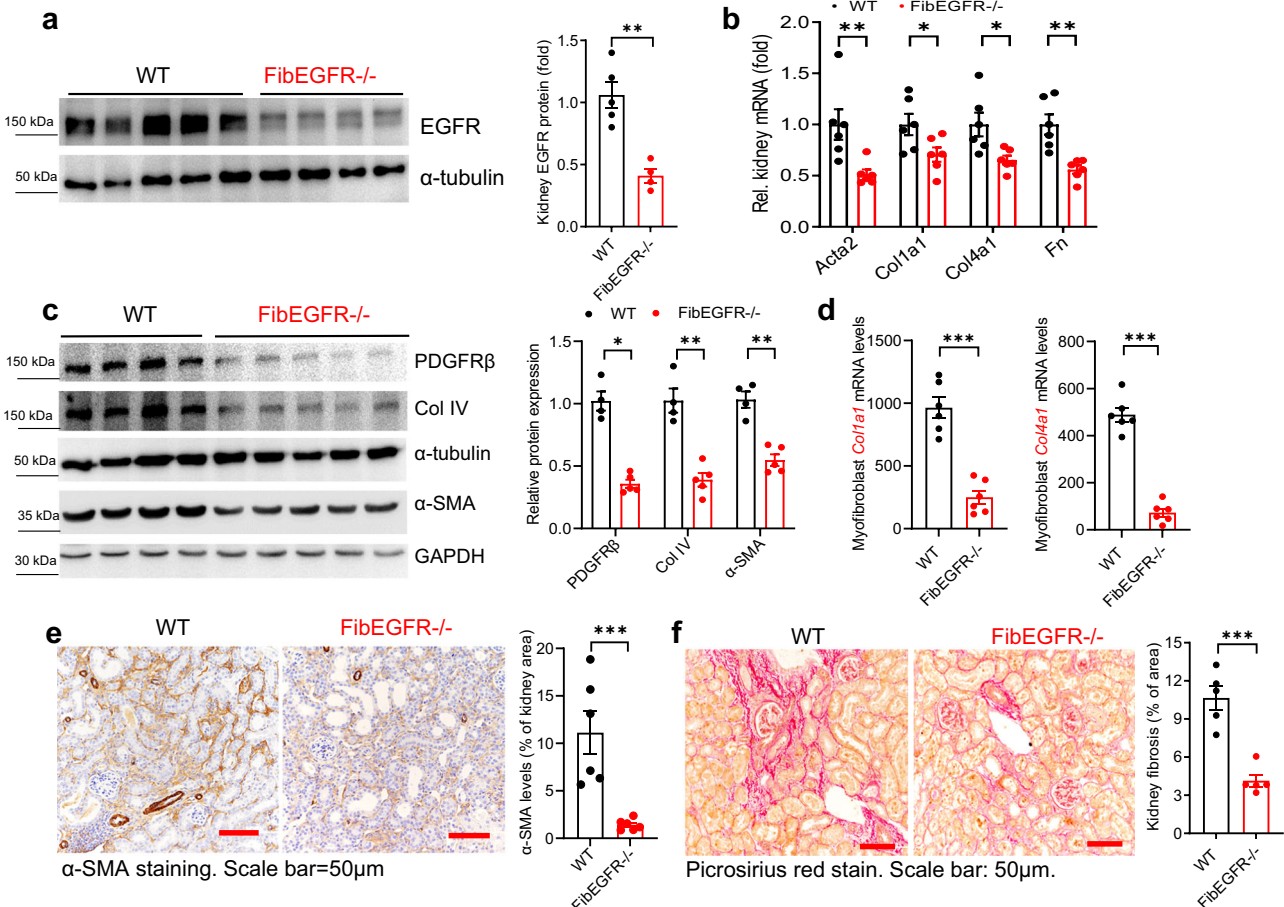

**Fig. 5 | Mice with selective fibroblast EGFR deletion developed less fibrosis after UUO. a** Immunoblotting indicated relatively lower total kidney EGFR levels in FibEGFR−/− mice at day 7 after UUO. $n = 4$ and 5. FibEGFR−/− mice exhibited less kidney fibrosis after UUO for 7 days as indicated by qPCR for kidney *Acta2*, *Col1a1*, *Col4a1* and *Fn* ($n = 6$) (**b**), immunoblotting for PDGFRβ, Collagen IV (Col IV), and α-

SMA ($n = 4$ and 5) (**c**), qPCR for *Col1a1* and *Col4a1* in isolated myofibroblasts ($n = 6$) (**d**), quantitative α-SMA immunostaining ($n = 6$) (**e**) and Picrosirius red staining ($n = 5$) (**f**). Data are means ± SEM, $*P < 0.05$, $**P < 0.01$, $***P < 0.001$, analyzed using two tailed Student's *t* test for all.

## Fibroblast EGFR played an essential role in fibroblast migration and fibroblast proliferation

Gene ontology analysis of the snRNA results indicated that the Fibroblast 2 and Myofibroblast 2 clusters were associated with aerobic respiration and oxidative phosphorylation and the Myofibroblast 2 cluster was also associated with intracellular and cell extracellular matrix organization, consistent with cells that were still relatively quiescent one day after UUO. The Fibroblast 1 cluster was associated with kidney development, the Myofibroblast 3 cluster with immune cell activation and differentiation and the Myofibroblast 4 cluster with cell proliferation. The Myofibroblast 5 cluster was associated with negative regulation of cell motility, consistent with decreased motility in the mature myofibroblasts (Fig. 8a). In addition, the Myofibroblast 1 cluster was also associated with decreased motility and migration. 60% of the total fibroblast population of FibEGFR−/− mice was in this cluster 3 days after UUO compared to only 15% in the WT (Figs. 4e, 8a).

Trajectory analysis indicated a continuous change in the RNA profile in WT fibroblasts from day 1 to day 3. Pseudotime analysis was performed to infer paths of cell-state transitions within the fibroblast clusters (Fig. 8b). In combination with GO terms, the predominance of Fibroblast 2 and Myofibroblast 2 clusters in the first day after UUO was consistent with their relative quiescence (Fig. 8c, d). Fibroblast 1 and Myofibroblast 1, 3 and 4 clusters, which were increased on day 3, may represent transdifferentiation and migration of pericytes as well as influx of fibrocytes. Fibroblast 1 and Myofibroblast 3 and 4 clusters

were associated with increased proliferation on day 3. These findings suggested that EGFR deletion in PDGFRβ+ cells, particularly on day 3 of UUO, primarily impacted the migration and proliferation of fibroblasts while only slightly affecting the differentiation and maturation of myofibroblasts.

In cultured murine fibroblasts, EGFR activation led to decreased mRNA levels of genes associated with myofibroblast differentiation and activation and fibrogenesis, including *Acta2*, *Col1a1*, *Col4a1*, *Fn*, *Pdgfrb*, *Tgfb1*, *Il11*, *Birc5*/survivin and *Itgb1*/integrin ß1 (Supplementary Fig. S9a–i). The observed decrease in kidney mCherry+ and α-SMA+ cells after UUO in FibEGFR−/− mouse kidney could be the result of decreased migration and/or proliferation of pericytes and fibroblasts. We used a Boyden Chamber assay to assess migration of cultured fibroblasts[36–38]. Administration of the EGFR ligand, HB-EGF led to marked fibroblast migration and proliferation, which was blocked by deletion of EGFR expression or the EGFR tyrosine kinase inhibitor AG1478 (Supplementary Fig. S10a, b). We took advantage of the endogenous mCherry fluorophore for our flow cytometry analysis to investigate whether EGFR deletion affected pericyte/fibroblast proliferation in vivo at day 3 after UUO, when quantitative Picrosirius red stain indicated that fibrosis had begun to increase in WT mice (Fig. 8e). Kidney mCherry+ fibroblasts/myofibroblasts were significantly lower in FibEGFR−/− mice than WT mice (Fig. 8f). We administered Click-iT™ Plus EdU Alexa Fluor™ 647 intraperitoneally 4 h before sacrifice to assess alterations in S

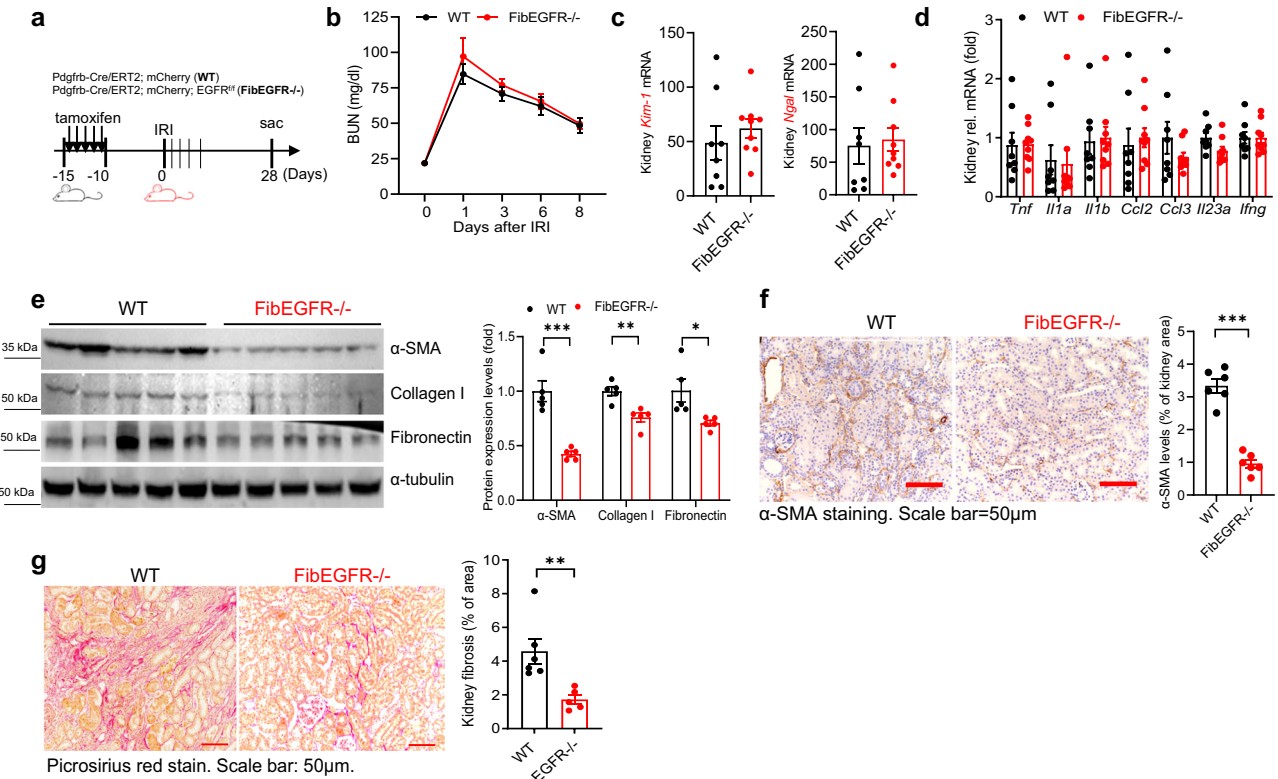

**Fig. 6 | Fibroblast EGFR deficiency did not affect kidney immune cell infiltration but attenuated development of kidney fibrosis after ischemic acute kidney injury. a** Schematic of experimental protocol. There was similar recovery between FibEGFR−/− mice and WT mice after ischemic AKI, as indicated by similar BUN decline (**b**), similar kidney injury as indicated by comparable transcripts of *Kim1* and *Ngal* (**c**), and similar kidney proinflammatory cytokines/chemokines (**d**). *n* = 8 and 9. FibEGFR−/− mice developed less kidney fibrosis 4 weeks after AKI as indicated by immunoblotting for α-SMA, collagen I and Fibronectin (*n* = 5) (**e**), quantitative α-SMA immunostaining (*n* = 6) (**f**) and Picrosirius red staining (*n* = 5 and 6) (**g**). Scale bar = 50 μm. Data are means ± SEM, \*\**P* < 0.01, analyzed using two-way ANOVA followed by Tukey's post hoc test for (**b**); two tailed Student's *t* test for (**c**–**g**).

phase mitosis. The percentage of proliferating fibroblasts (Edu⁺ mCherry⁺) was also markedly lower in FibEGFR−/− mice than in WT mice (Fig. 8g). Immunofluorescence confirmed that the number of PCNA⁺α-SMA⁺ double positive cells was also significantly lower in kidneys of FibEGFR−/− mice compared to WT mice (Fig. 8h). Of note, PCNA⁺ tubular cells were also observed in in both WT and FibEGFR−/− mice.

### EGFR signaling opposed TGF-β signaling to induce fibroblast differentiation into myofibroblasts

In cultured murine fibroblasts, administration of TGF-β did not itself affect fibroblast proliferation nor did it alter the proliferation induced by HB-EGF (Supplementary Fig. S10a). Although TGF-β alone did not itself stimulate or inhibit fibroblast migration, it did inhibit HB-EGF-induced migration (Supplementary Fig. S10b–d).

Simultaneous TGF-β administration also overcame EGFR activation-mediated inhibition of genes associated with myofibroblast differentiation and activation and fibrogenesis (Supplementary Fig. S9a–i). TGF-β-induced increases in *Acta2*, *Col1a1*, *Fn*, and *Il11* transcripts were further enhanced by inhibition of EGFR tyrosine kinase activity with AG1478 (Supplementary Fig. S9j). EGF inhibition of *Smad3* expression was also prevented by TGF-β (Supplementary Fig. S9k).

As noted, TGF-ß is an important mediator for myofibroblast transformation. In response to UUO, whole kidney levels of *Tgfb1*, *Tgfb2*, and *Tgfb3* transcripts progressively increased in WT mice (Fig. 9a), which correlated with timing of progressive increases in fibrosis. SMAD3 is a key downstream signaling molecule for TGF-ß-mediated fibrogenesis. At day 3 after UUO, not only were α-SMA+

myofibroblasts decreased in FibEGFR−/− mice compared to WT mice, but there was also a lower percentage of phospho-SMAD3-positive myofibroblasts (Fig. 9b), consistent with previous studies, further indicating the important role that EGFR plays in fibroblasts in TGF-ß-mediated development of kidney fibrosis[12,39–41].

In the snRNA analysis, TGF-ß family members had greatest enrichment in the MF2 fraction on day one but also had increased expression in Fib2 and MF3. In contrast, EGF family members had greatest enrichment in the Fib2 fraction on day one (Fig. 9c). Analysis of ligand-receptor interactions demonstrated that *Hbegf* from the *EGF* family interacted with EGFR in subclusters in Fib1, MF3, and MF4, all of which were associated with proliferation on day 3 (Fig. 9d). Of note, HB-EGF is also a ligand for ErbB4. We have previously reported that global deletion of ErbB4 resulted in increased tubulointerstitial fibrosis in response to UUO or ischemic injury[42]. Both HB-EGF and ErbB4 interaction and Nrg1 (an ErbB4 ligand) and ErbB4 interaction were also evident in Fibroblast 1 and Myofibroblast 3 and 4 clusters, suggesting a potential role to counteract the effects of EGFR.

Ligands and receptors detected with a positive average fold enrichment were paired to elucidate intercellular signal transduction networks. Compared to the Fibroblast 1 or Fibroblast 2 cluster, all five Myofibroblast clusters had increased gene expression for ligands that can theoretically interact with receptors on kidney epithelium, especially proximal tubule and thick ascending limb, as well as endothelium, pericytes and innate and adaptive immune cells (Supplementary Fig. S11). Interestingly, at day 3, there was less evidence of ligands from intrinsic kidney cells interacting with the receptors on the fibroblast or myofibroblast clusters.

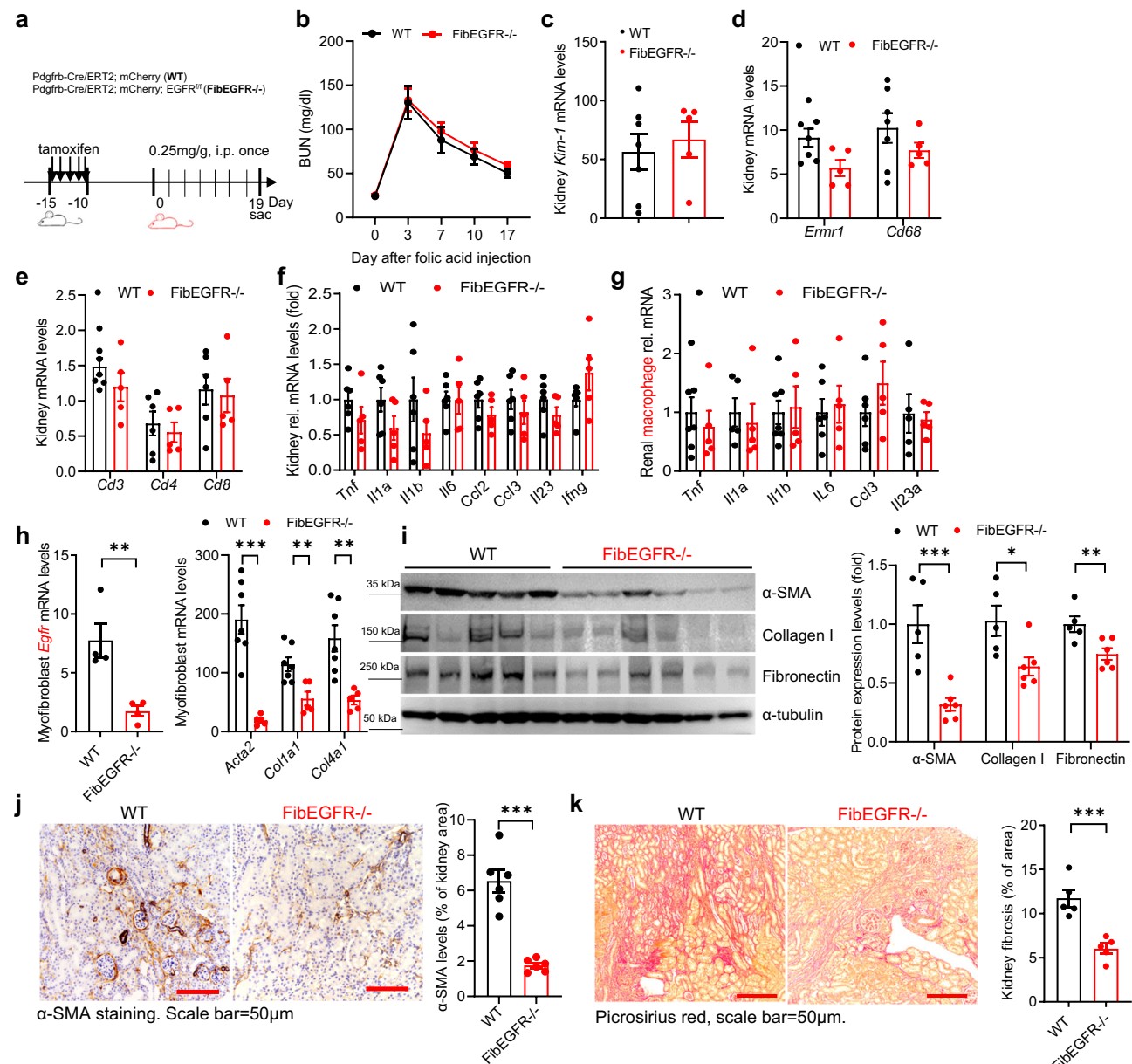

**Fig. 7 | Fibroblast EGFR deficiency did not affect kidney immune cell infiltration but attenuated development of kidney fibrosis in folic acid nephropathy.**
**a** Schematic of experimental protocol. Similar kidney functional recovery (*n* = 6 and 8) (**b**), kidney injury (*n* = 5 and 7) (**c**), kidney immune cell infiltration (*n* = 5 and 6 and 7) (**d**, **e**), and proinflammatory cytokines/chemokines (*n* = 5 and 6) (**f**) were observed in FibEGFR−/− mice and WT mice with folic acid nephropathy.
**g** Proinflammatory cytokine/chemokine transcripts were comparable in isolated kidney macrophages between FibEGFR−/− mice and WT mice after folic acid administration. *n* = 5 and 6 and 7. **h** Myofibroblasts isolated from FibEGFR−/− mice

after folic acid administration expressed markedly lower mRNA levels of *Egfr* (*n* = 4), *Acta2* (*n* = 5 and 7), *Col1a1* (*n* = 5 and 7), and *Col4a1* (*n* = 5 and 7). FibEGFR−/− mice developed less kidney fibrosis in response to folic acid nephropathy as indicated by immunoblotting for α-SMA, collagen I and Fibronectin (*n* = 5 and 6) (**i**), quantitative α-SMA immunostaining (*n* = 6) (**j**) and Picrosirius red staining (*n* = 5) (**k**). Scale bar = 50 μm. Data are means ± SEM, **P* < 0.01, ***P* < 0.001, analyzed using two-way ANOVA followed by Tukey's post hoc test for (**b**); two tailed Student's *t* test for (**c**–**k**).

## EGFR was expressed in myofibroblasts in fibrotic human kidneys

We also used available scRNAseq datasets available to assess mRNA expression EGFR and its ligands in mouse ischemic injury[34]. As indicated in Supplementary Fig. S12, EGFR mRNA increased in fibroblasts and tubular epithelial cells as early as 4 h after ischemic injury. In addition, the mRNA levels of HB-EGF and AREG as well as ADAM17 also increased at 4 h after ischemic injury (Supplementary Fig. S12). We also assessed EGFR mRNA expression in fibroblast sub-populations from published human CKD kidney scRNAseq[43]. As indicated in Figure Supplementary Fig. S13, fibroblasts and myofibroblasts from human CKD kidneys expressed high levels of EGFR mRNA. In samples of

human kidneys without fibrosis, α-SMA immunofluorescence was confined to pericytes, and EGFR was predominantly expressed in tubules. In contrast, in fibrotic kidneys, EGFR colocalized to α-SMA+ myofibroblasts in the interstitium (Fig. 10a). The etiologies of the kidney injuries are indicated in the legend to Fig. 10.

## Discussion

The current studies provide evidence for a heretofore undescribed and important role for EGFR in kidney fibroblasts and pericytes as a specific initiator of interstitial fibrosis in response to kidney injury. Previous studies had indicated that global genetic or pharmacologic blocking of

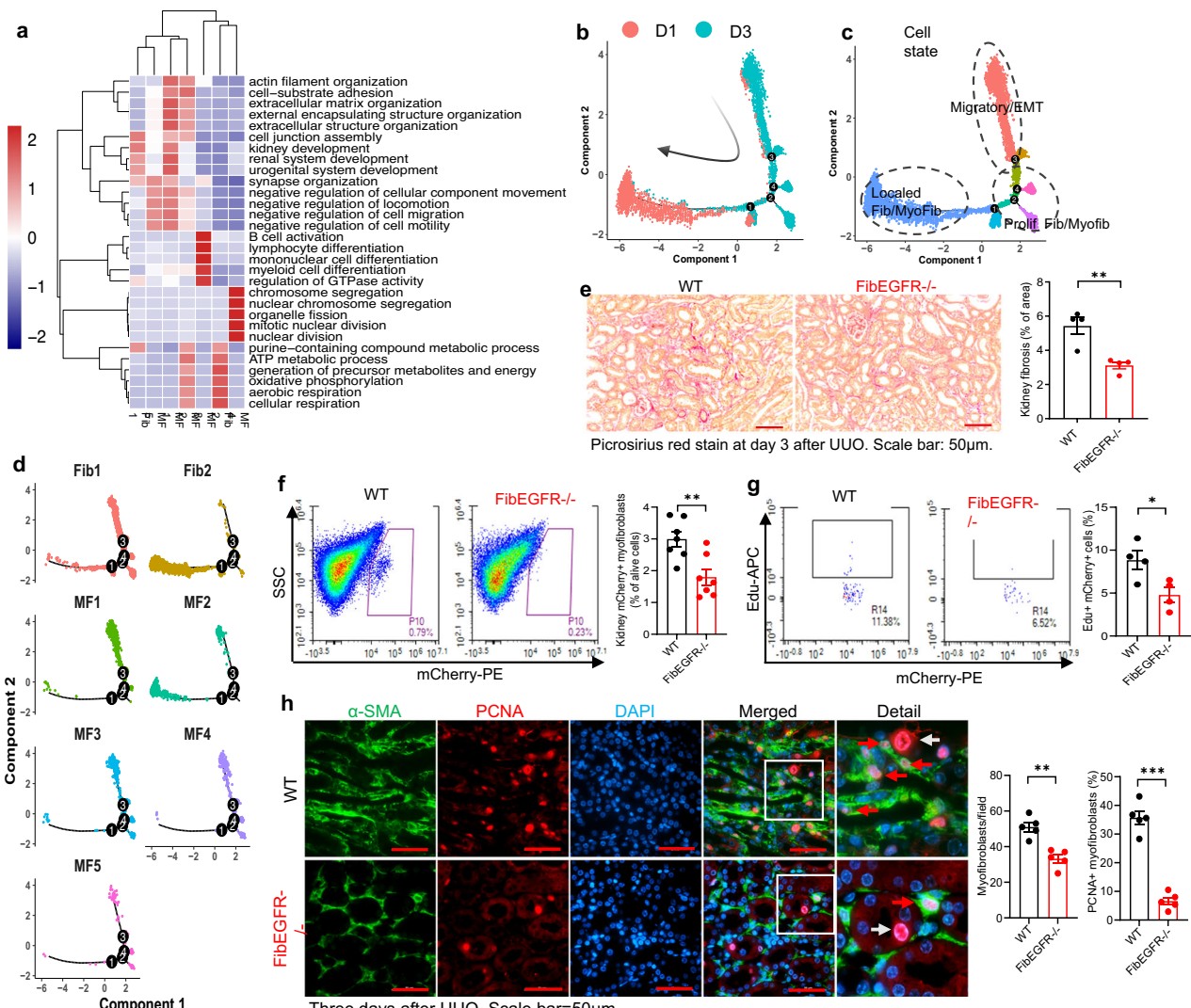

**Fig. 8 | Fibroblast EGFR deficiency led to less fibroblast proliferation in response to unilateral ureteral obstruction (UUO).** **a**–**d** snRNAseq analysis indicated characteristics of fibroblast and myofibroblast subclusters following unilateral ureteral obstruction (UUO). The R package CellChat was used, and CellChatDB is the built-in database in CellChat without a separate version. **a** Gene Ontology (GO) Biological Process analysis for fibroblast subclusters based on the top marker genes for each cluster suggested that Fib1 cluster was associated with kidney development, MF3 cluster with immune cell activation and differentiation, and MF4 cluster with fibroblast proliferation. **b** Trajectory analysis showed a continuous change in RNA profile in WT fibroblasts from day 1 to day 3 after UUO injury. **c** Pseudotime analysis was performed to infer paths of cell-state transitions within the fibroblast clusters. **d** In combination with GO terms, Fib2/MF2 dominated the first day in the UUO model, probably due to the activation and differentiation of local fibroblast cells in the kidney. Fib1/MF3 may arise from the

transdifferentiation of renal cells after injury or some cells from migration, and these fibroblast subclusters were more inclined to activate immune cells and promote differentiation and to participate in cross-talk with immune cells. MF4 was associated with fibroblast proliferation on tday three in the UUO model. **e** FibEGFR −/− had less kidney fibrosis compared to WT mice at day 3 after UUO. n = 4. Scale bar = 50 μm. Both the percentage of mCherry+ fibroblasts/myofibroblasts in total kidney cells (n = 7) (**f**) and the proliferation rate of fibroblasts/myofibroblasts (EdU+ mCherry+ double positive cells) (n = 4) (**g**) were markedly lower in FibEGFR−/− mice than WT mice. **h** Immunofluorescent staining showed fewer kidney PCNA+ α-SMA+ double positive cells (red arrows, proliferating fibroblasts/myofibroblasts) in FibEGFR−/− mice compared to WT mice. n = 5. Of note, PCNA+ positive cells were also observed in tubular cells in both WT and FibEGFR−/− mice. Scale bar = 50 μm. Data are means ± SEM, **P < 0.01, **P < 0.001, analyzed using two tailed Student's *t* test for all.

the EGFR signaling pathway inhibited kidney fibrosis while stimulating kidney EGFR activity promoted kidney fibrosis[11–15,17,30]. However, these studies did not determine the mechanism by which EGFR itself mediated fibrosis since EGFR activation has not been shown to directly stimulate a profibrotic program in fibroblasts or pericytes. In the present studies we deleted EGFR in kidney pericytes and resident fibroblasts and found significant inhibition of the number of interstitial fibroblasts/myofibroblasts and a subsequent decrease in interstitial fibrosis in response to a variety of experimental insults known to incite kidney fibrosis (Supplementary Fig. S14). Fibroblast/pericyte EGFR activation did not promote myofibroblast transformation but instead mediated increased proliferation and migration to sites of injury. Since

fibroblast EGFR activation does not itself induce myofibroblast differentiation, subsequent activation by TGF-β or other profibrotic agonists is necessary for their differentiation into myofibroblasts. However, without the crucial initial steps provided by EGFR activation, the ultimate development of tubulointerstitial fibrosis is severely impaired (Fig. 10b).

EGFR ligands exist as membrane-associated precursors that are cleaved to form their active ligands by metalloproteinases, especially ADAM17 (TACE). Kefaloyianni et al. reported that whole kidney ADAM17, EGFR, and EGFR ligands, including HB-EGF, TGF-α, AREG and EREG, all increased following ischemic injury[30], consistent with the current results indicating similar progressive increases in

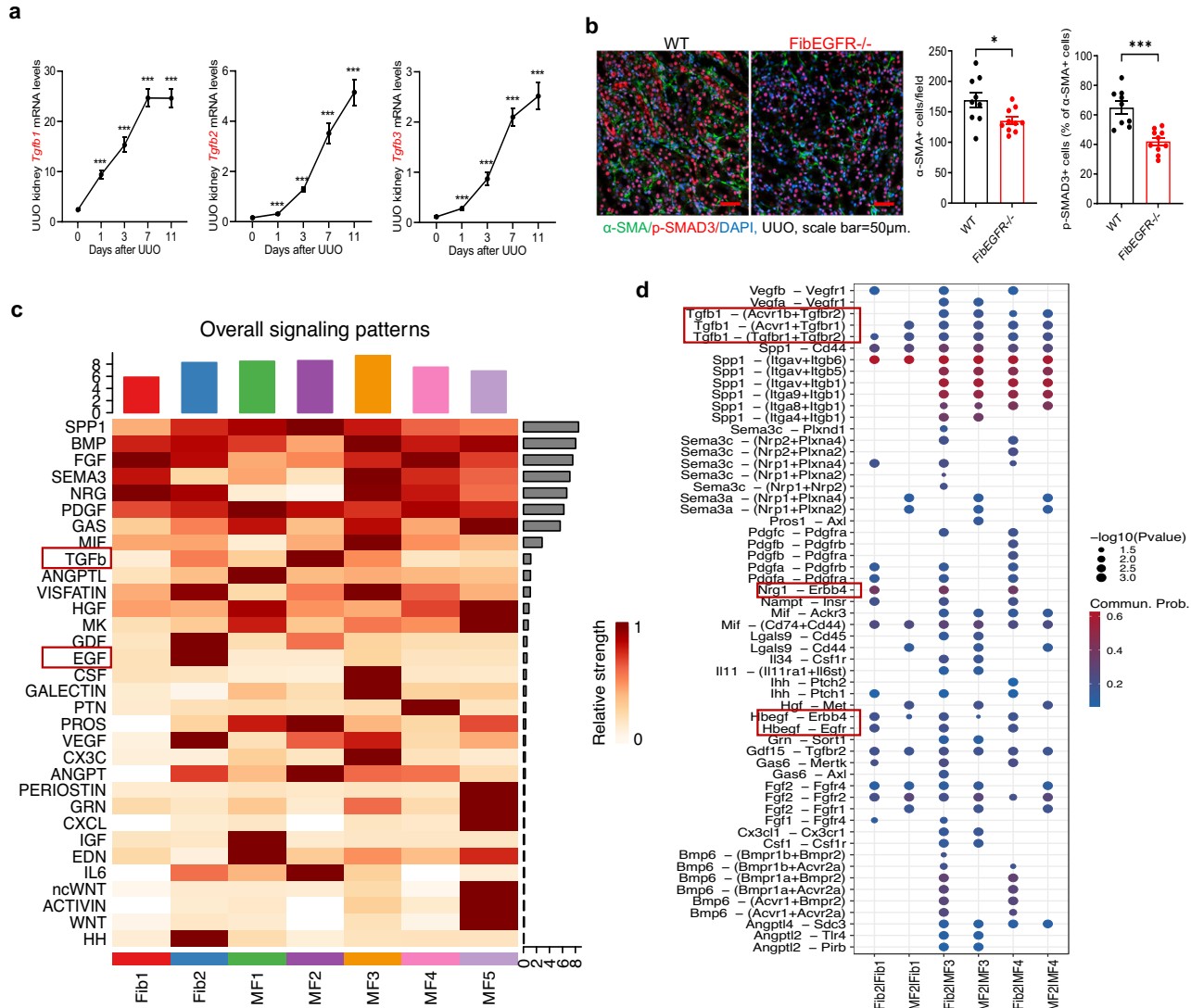

**Fig. 9 | Mice with fibroblast EGFR deletion had attenuated TGF-β signaling in fibroblasts/myofibroblasts. a** Kidney transcripts of *Tgfb1*, *Tgfb2*, and *Tgfb3* increased progressively after UUO in WT mice after UUO. *n* = 7 and 8. **b** Both the density of PDGFRβ+ fibroblasts/myofibroblasts and p-SMA3⁺PDGFRβ⁺ double positive cells were markedly lower in FibEGFR−/− mice than WT mice at day 7 after UUO. *n* = 9 and 10. Scale bar = 50 μm. **c, d** snRNA analysis shows interactions of TGF-β signaling and EGFR signaling in fibroblasts/myofibroblasts (Fib/MF) on day one following unilateral ureteral obstruction (UUO). **c** Heat maps shows that secreted phosphoprotein 1 (SPP1), bone morphogenetic protein (BMP), fibroblast growth

factor (FGF), and neuregulin (NRG) were the major ligands from Fib/MF cells across all subgroups. TGF-ß family members had the greatest enrichment in the MF2 fraction while EGF family members had the greatest enrichment in the Fib2 fraction on day one. **d** Analysis of ligand-receptor interactions indicated HB-EGF interactions with EGFR and Erbb4 as well as Nrg1 interaction with Erbb4 in subclusters in Fib1, MF3, and MF4 while TGF-β1 interacted with TGF-ß receptors in all fibroblast/myofibroblast subtypes. Data are means ± SEM, *P < 0.05, ***P < 0.001, analyzed using 2-way ANOVA followed by Tukey's post hoc test for (**a**), and 2 tailed Student's *t* test for (**b**).

response to UUO. Published scRNA sequences also showed increased EGFR, HB-EGF, AREG, and ADAM17 mRNA expression after mouse ischemic kidney injury model as well as high EGFR mRNA in fibroblasts and myofibroblasts in CKD kidneys (Supplementary Figs. S12, S13). Either ADAM17 hypomorphic mice or mice with proximal tubule selective deletion of ADAM17 had less kidney interstitial fibrosis after either ischemic injury or in a UUO model[30]. iRhoms, members of the Rhomboid intramembrane protein family, are essential mediators of ADAM17 activation. In the current study, we found that deletion of iRhom2 inhibited tubulointerstitial fibrosis after ischemic injury.

Although iRhom1 is ubiquitously expressed, iRhom2 was previously reported to be predominantly expressed in cells of myeloid lineage under basal conditions[18,44]. However, we found that following kidney injury, total kidney mRNA expression of both iRhom1 and iRhom2 increased dramatically. We also detected iRhom2 protein expression colocalized with α-SMA in interstitial cells following UUO.

In addition to association with ADAM17 at the plasma membrane, iRhom2 is necessary for translocation of ADAM17 from the endoplasmic reticulum to the Golgi. Interestingly, in the α-SMA+ myofibroblasts, the greatest iRhom2 immunofluorescence was detected in the paranuclear region, consistent with localization to Golgi and ER (Fig. 1d).

It was striking that although there was no difference in immunoreactive EGFR colocalization with myofibroblasts between wild type and iRhom2−/− mice, the colocalization of phospho-EGFR in the myofibroblasts was significantly decreased in iRhom2−/− mice compared to WT mice 7 days after UUO (Fig. 1j). In addition, we found that there was increased EGFR expression in kidney interstitial fibroblasts from both humans and experimental animals with tubulointerstitial fibrosis. EGFR activation has previously been shown to induce a feedforward response to further increase EGFR expression[14]. These findings suggested that EGFR activation in myofibroblasts was involved in the

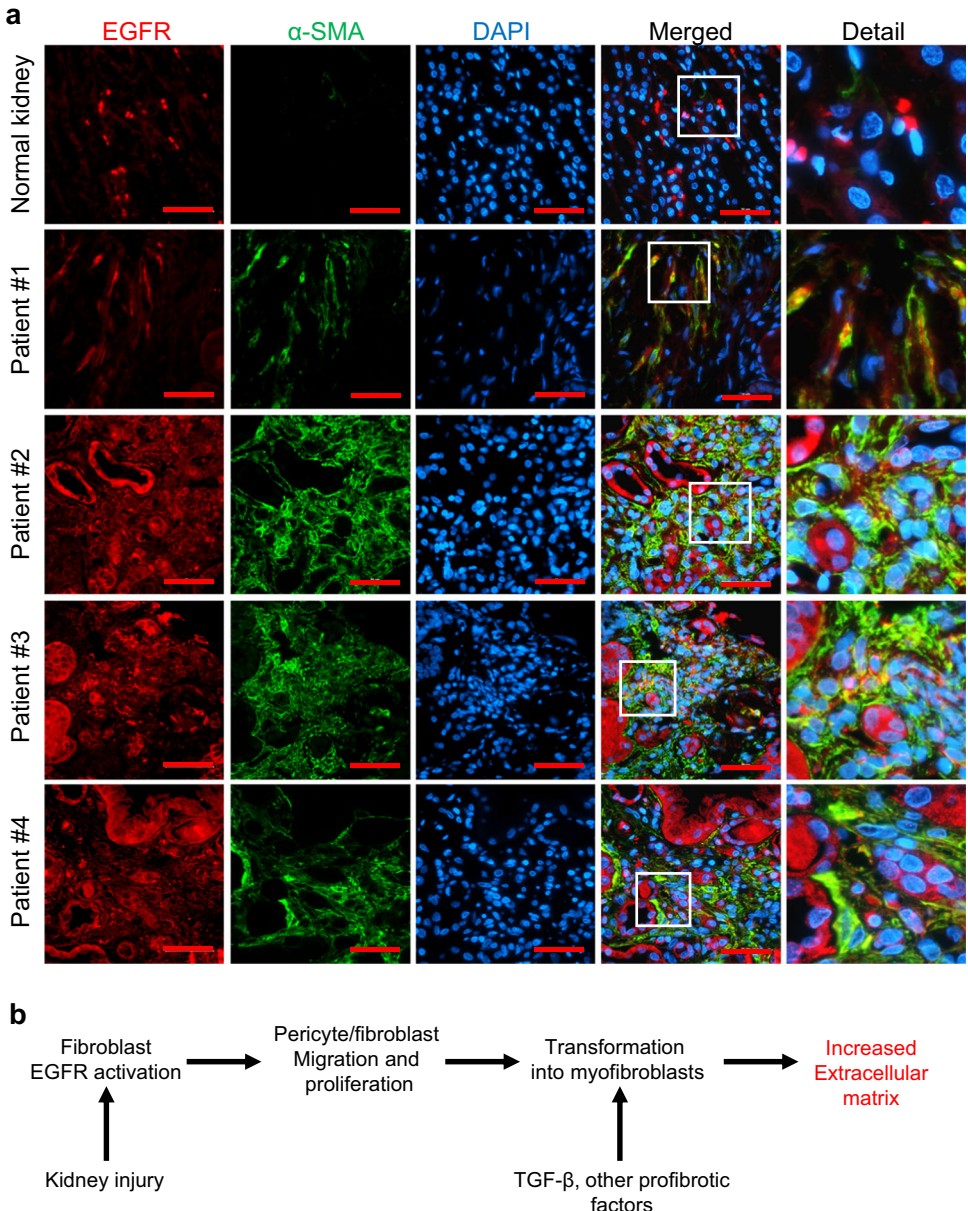

**Fig. 10 | EGFR was highly expressed in kidney myofibroblasts from both male and female patients. a** In normal human kidney wdithout fibrosis, EGFR was expressed in the tubules. In fibrotic human kidneys, EGFR was also expressed in myofibroblasts (colocalization with α-SMA) in the interstitium. Patient #1 was a middle-aged African American man with hypertension and HIVAN; Patient #2 was an elderly women with hypertension; patient #3 was a middle-aged man with hypertension and history of IV drug abuse; and patient #4 was a middle-aged old woman with hypertension and possible chronic pyelonephritis. Therefore, EGFR expression was elevated in kidney myofibroblasts in fibrotic kidneys from both genders. Scale bar = 50 μm. **b** Summary of EGFR activation in development of kidney fibrosis after injury.

development of tubulointerstitial fibrosis in response to injury. However, the identity and source(s) of the ligands involved are beyond the scope of the current studies since it is likely that more than one EGFR ligand may be involved. However, by taking an alternative approach to investigate the target cells, we are the first to determine that direct activation of EGFR on fibroblasts mediates the observed effect of EGFR to promote tubulointerstitial fibrosis.

EGFR activation is well known to induce cell dedifferentiation and proliferation, and our results indicate that these properties mediate its essential function in the development of interstitial fibrosis in response to kidney injury. Selective EGFR deletion decreased markers of proliferation, Ki 67 and EdU, in PDGRß positive cells after UUO. In addition, the total number of myofibroblasts (α-SMA + cells) was decreased. EGFR activation increased both the proliferation and the migratory ability of

cultured mouse fibroblasts and inhibited mRNA markers of fibrosis (α-SMA, collagen 1, fibronectin). Of note, simultaneous administration of TGF-ß was able to induce markers of myofibroblast differentiation, which were further increased with simultaneous administration of an EGFR inhibitor. Since TGF-ß itself did not induce either fibroblast proliferation or migration, these results suggest that the initial step is activation of EGFR in PDGFRß-positive cells, leading to migration and proliferation, with subsequent inhibition of these processes and differentiation to myofibroblasts and ultimate development of tubulointerstitial fibrosis as profibrotic cytokine levels increase, timing which would be consistent with the time course of progressive expression of TGF-ß mRNA after injury observed in the current studies.

Selective deletion of EGFR in the PDGFRß-positive cell population inhibited fibrosis not only in the UUO model, but also in response to

ischemic or toxic (folic acid, adenine) injury. Of note, in the latter models, the inhibition of interstitial fibrosis was not accompanied by any alteration of inflammatory cytokines, tubule injury markers or functional recovery. These findings are consistent with previous studies by Kefaloyianni et al. that global or proximal tubule deletion of ADAM17 or proximal tubule deletion of amphiregulin did not alter functional recovery from ischemic kidney injury[16,30].

snRNAseq delineated 7 distinct subclusters of fibroblasts and myofibroblasts in the UUO kidneys. These studies were designed to compare the early events following acute kidney injury, as indicated by the fact that there were still relatively few mature myofibroblasts present on day 3 after UUO. On day one after UUO, the majority of these cells were still in subclusters in a relatively differentiated and quiescent state, as indicated by evidence of aerobic respiration and oxidative phosphorylation, and there were equivalent numbers of fibroblasts in wild type and FibEGFR−/− mice in these subclusters. In contrast, on day 3, the total number of fibroblasts increased markedly between day 1 and day 3 post UUO in wild type kidneys but failed to increase in the FibEGFR−/− kidneys, confirming the important role that EGFR activation plays in the activation of fibroblasts as they transition to myofibroblasts. The snRNAseq results also confirmed that EGFR deletion in the PDGFRß+ cells markedly decreased the migration and proliferation of fibroblasts with relatively minimal effect on the differentiation and maturation of myofibroblasts.

Gene expression of EGFR ligand family members in the PDGFRß+ cell population was highest on day one and was primarily in the differentiated fibroblast Fib2 cluster. Analysis of ligand-receptor interactions indicated that HB-EGF from the EGF family interacted with EGFR in subclusters on day 3 with the subclusters associated with proliferation and migration. TGF-ß gene family members had greatest enrichment in the early myofibroblast subcluster on day one but had persistent expression in subclusters that predominated on day 3, and TGF-ß interacted with TGF-ß receptors in all fibroblast and myofibroblast subtypes. Although our results do not rule out paracrine activation, these findings suggested that HB-EGF and TGF-ß1 may facilitate the migration, proliferation, and subsequent differentiation of fibroblasts/myofibroblasts in an autocrine manner.

It was noteworthy that in addition to EGFR (ErbB1), another ErbB family member, ErbB4 is expressed in adult mammalian kidney. We have previously reported that global deletion of ErbB4 resulted in increased tubulointerstitial fibrosis in response to acute kidney injury, suggesting a potential role to counteract the effects of EGFR[42]. In addition to EGFR, HB-EGF is known to be a ligand for ErbB4, and gene analysis also suggested interaction with ErbB4 in the proliferating Fibroblast1 and Myofibroblast 3 and 4 clusters (Fig. 9b). These clusters also expressed increased mRNA for another ErbB4 ligand, NRG1 (neuregulin 1). These findings suggest that ErbB4 activation in fibroblasts might serve to counter the effects of EGFR activation, but further studies will be required to investigate this hypothesis.

Although the current studies are focused upon the development of interstitial fibrosis following acute kidney injury, it is likely that the findings may also provide insight into development of fibrosis in other organs and in other conditions. Fibrosis of the pancreas is a poor prognostic sign in both chronic pancreatitis and pancreatic cancer[45], and EGFR activation of the fibroblast-like pancreatic stellate cells has been implicated in development of pancreatic fibrosis in that organ[46]. EGFR activation has also been implicated in development of fibrosis in both lung and liver[47,48].

In summary, these studies demonstrate the important role of EGFR activation of the PDGFRß- positive population of pericytes and fibroblasts in development of kidney interstitial fibrosis in a variety of experimental models of tubulointerstitial fibrosis and elucidate the underlying mechanisms. EGFR activation does not mediate myofibroblast transformation per se but instead directs the initial pericyte/fibroblast migration and proliferation prior to subsequent myofibroblast transformation by TGF-ß or other profibrotic factors. Therefore, these studies provide new insights into our understanding of the mechanisms underlying development of interstitial fibrosis in response to kidney injury.

## Methods
### Animals
All animal experiments were performed in accordance with the guidelines and with the approval of the Institutional Animal Care and Use Committee of Vanderbilt University. iRhom2−/− mice (C57BL/6J) were purchased from The Jackson Laboratory (#044040).EGFR floxed (EGFR^f/f) mice were generated by flanking exon 3 of the EGFR gene with two LoxP sites, as described previously[49]. iRhom2 knockout mice (strain#044040, C56BL/6), PDGFRβ-P2A-CreER^T2 mice (Strain #: 030201, C56BL/6), and R26 LSL H2B mCherry (Rosa26-CAG-LSL-H2B-mCherry mice (Strain #023139) were all purchased from The Jackson Laboratory. We generated mice with selective EGFR deletion in pericytes/fibroblasts/myofibroblasts with mCherry (PDGFRβ-Cre/ERT2; mCherry; EGFR^f/f, FibEGFR−/−) and corresponding wild type (PDGFRβ-Cre/ERT2; mCherry, WT). Male 8–12 weeks old mice were used at the initiation of experiments and genotyped with PCR before and after the study was completed. The primers used for genotyping in the current studies are listed in Supplementary Table S1. Mice were euthanized with CO2 method plus cervical dislocation to ensure the death of the animals.

### Ischemic acute kidney injury (AKI), unilateral ureteral obstruction (UUO), folic acid nephropathy and adenine nephropathy
Both PDGFRβ-Cre/ERT2; mCherry and PDGFRβ-Cre/ERT2; mCherry; EGFR^f/f mice were treated with tamoxifen at 160 mg/kg via intraperitoneal injection for 5 consecutive days (Sigma, T6648, in corn oil). Ten days after the last tamoxifen injection, these animals were used for experiments. The mice were anesthetized with isoflurane with Precision vaporizer for all surgeries and before sacrifice. Ischemia-reperfusion-induced AKI was carried out as previously described[28,50]. Briefly, the animals were uninephrectomized, immediately followed by unilateral ischemia-reperfusion with kidney pedicle clamping for 32 min. For UUO, the left ureter was ligated for the indicated days. Folic acid nephropathy was induced by a single intraperitoneal injection of folic acid at a dose of 0.25 mg/g (F7876, Sigma-Aldrich) and the animals were sacrificed after 19 days. Adenine nephropathy was induced by feeding the mice with a diet containing 0.25% adenine throughout the experiments (Envigo RMS Inc. Catalog# TD. 140007).

### Antibodies and reagents
The information of all antibodies used is listed in Supplementary Table S2.

### In vitro migration assay and cell proliferation assay
Migration assays were performed as previously described[51]. Mouse fibroblasts (100,000 cells) were seeded onto the top chamber of a 24-well PET membrane (8 mm pore size). Cells translocated to the lower chamber in response to exposure in the lower chamber of medium (vehicle), HB-EGF (50 ng/ml, cat# E4643, Sigma-Aldrich), HB-EGF plus an EGFR inhibitor (AG1478, 10 μM, LC laboratories) or TGF-β (5 ng/ml, cat# 7754BH005CF, R&D Systems™) in the presence or absence of 5% serum for 16 h. Cells in the upper chamber were removed with a cotton swab, and the filters were fixed with 70% ethanol and stained with 2% crystal violet. Filters were photographed on a Leica DMi1 microscope, and the total cell number was counted. Similar experiments were repeated in mouse fibroblasts with EGFR deletion. For the cell proliferation assay, fibroblasts were treated with HB-EGF (50 ng/ml), TGF-β (5 ng/ml) or a combination of both for 16 h. EdU (10 μM) was added to the culture medium 2 h before harvesting. Cells were permeabilized and followed by the Click-iT EdU reaction (Click-It Plus EdU flow

cytometry assay kit, Thermo Fisher) according to the manufacturer's protocols and analyzed by flow cytometry described below.

## Establishment of ErbB1 (EGFR) knockout cell lines

The crRNA DNA sequence (5′-GACCGCGAGAACCA CACTGC-3′) for knockout of ErbB1 (EGFR) was inserted in the BbSI sites of pX330-U6-Chimeric-BB-CBh-hSpCas9 (Addgene) (HspCas9-ErbB4 E1b and HspCas9-ErbB1-E1a). $5 \times 10^5$ immortalized mouse fibroblasts in a well of a 6 well plate were transfected with 1 μg of HspCas9-ErbB1-E1a, 1 μg of PB-pCMV-MCS-EF1a-GFPT2APURO (System Biosciences) and 0.5 μg of HAPB plasmids with Lipofectamine LTX and Plus (Invitrogen). After 24 h, fibroblasts were split to 96 well plates and selected with 3 μg/ml of puromycin. Single colonies were selected and expanded and ErbB1 protein was examined via western blot. Genomic DNA was extracted from fibroblast cell lines with western blot confirming knockout of ErbB1 using a DNeasy Blood and Tissue kit (Qiagen). Gene fragments flanking the crRNA sequences of ErbB1 (ERBB1 F 5′-AGTCCCGACCC GAGCTAAC-3′, ERBB1 R 5′- TGGGACACGCCCTTACCTTTC-3′) were amplified using Q5 high-fidelity DNA polymerase (New England Bio-Labs) and purified with a NucleoSpin Gel and PCR Clean-up kit (Macherey-Nagel). Purified gene fragments were sequenced to confirm the knockout of ErbB1 (Genewiz).

## Flow cytometry analysis

Cell proliferation in the S phase was evaluated using Click-iT™ Plus EdU Alexa Fluor™ 647 Flow Cytometry Assay Kit (Invitrogen™, C10634), which was administrated via intraperitoneal injection 4 h before sacrifice according to the manufacturer's protocol. For detection of proliferation, cells were gated on CD45 negative, CD31 negative, and mCherry positive cells before counting the number of events that were both mCherry and EdU positive. Kidney single cell suspensions were prepared according to previous reports[51]. Cells were fixed and permeabilized with Cytofix/Cytoperm Solution (BD Biosicences, Cat# 554722) for 20 min and washed with Perm/Wash buffer (BD Biosicences, Cat# 554723) and then incubated in 2.5 μg/ml Fc blocking solution, centrifuged ($300 \times g$, 10 min, 4 °C) and resuspended with FACS buffer. Cells were stained for 60 min at 4 °C with fluorescent conjugated antibodies against different cell marker antigens (Table S2) or isotype control. A total of 50,000–1,00,000 cells were acquired by scanning using NovoCyte Quanteon Flow Cytometer Systems. Cell debris and dead cells were excluded from the analysis based on scatter signals and use of Zombie Violet™ Fixable Viability Kit (Biolegend, Cat# 423114).

## Isolation of kidney fibroblasts/myofibroblasts

Kidney fibroblasts/myofibroblasts were isolated using PDGFRβ antibody (Sigma-Aldrich, Cat# 06-498-I) plus anti-rabbit IgG microbeads (cat# 130-048-602, Milteni Biotec Auburn, CA) and MACS columns (130-110-443, Milteni Biotec Auburn, CA) following the manufacturer's protocol.

## Quantitative immunofluorescence/immunohistochemistry staining

Kidney tissue was immersed in fixative containing 3.7% formaldehyde, 10 mM sodium m-periodate, 40 mM phosphate buffer, and 1% acetic acid. The tissue was dehydrated through a graded series of ethanols, embedded in paraffin, sectioned (4 μm), and mounted on glass slides. Antigen retrieval in the deparaffinized sections was performed with citrate buffer by microwave heat for 10 min and the slides were then blocked with 10% normal donkey serum for 1 h followed by incubation with primary antibodies overnight at 4 °C. For double immunofluorescence staining, the sections were incubated in two rounds of staining overnight at 4 °C. Anti-rabbit or mouse IgG-HRP were used as secondary antibodies (Santa Cruz). Each round was followed by tyramide signal amplification with the appropriate fluorophore (Alexa

Flour 488 tyramide, Alexa Flour 647 tyramide or Alexa Flour 555 tyramide, Tyramide SuperBoost Kit with Alexa Fluor Tyramides, Invitrogen) according to its manufacturer's protocols. DAPI was used as a nuclear stain. Sections were viewed and imaged with a Nikon TE300 fluorescence microscope and spot-cam digital camera (Diagnostic Instruments), followed by quantification of cells/field using Image J software (NIH, Bethesda, MD). IOD were calculated in more than 30 fields per mouse or 10 fields per cell slide and expressed as arbitrary units or percentage of per field by two independent investigators.

## Immunoblotting analysis

Whole kidney tissue was homogenized with lysis buffer containing 10 mmol/l Tris–HCl (pH 7.4), 50 mmol/l NaCl, 2 mmol/l EGTA, 2 mmol/l EDTA, 0.5% Nonidet P-40, 0.1% SDS, 100 μmol/l Na3VO4, 100 mmol/l NaF, 0.5% sodium deoxycholate, 10 mmol/l sodium pyrophosphate, 1 mmol/l PMSF, 10 μg/ml aprotinin, and 10 μg/ml leupeptin and centrifuged at $15,000 \times g$ for 20 min at 4 °C. The BCA protein assay kit (Thermo Scientific) was used to measure the protein concentration. Immunoblotting was performed as previously described[52,53] and quantitated with Image J software.

## Quantitative PCR

Total RNAs from kidneys or cells were isolated using Trizol® reagent (Invitrogen). SuperScript IV First-Strand Synthesis System kit (Invitrogen) was used to synthesize cDNA from equal amounts of total RNA from each sample. Quantitative RT-PCR was performed using TaqMan real-time PCR (7900HT, Applied Biosystems). The Master Mix and all gene probes were also from Applied Biosystems. The primers used for qPCR in the experiments were presented as Supplementary Table S3. Realtime PCR data were analyzed using the 2-ΔΔCT method to determine the fold difference in expression. Generally, 500 ng mRNA, measured by Nanodrop 2000, was obtained from isolated kidney tissue myeloid cells.

## Picrosirius red stain

Picrosirius red stain was performed according to the protocol provided by the manufacturer (Sigma, St. Louis, MO, USA).

## Measurement of BUN

BUN was measured using a Urea Assay Kit (BioAssay Systems, Hayward, CA).

## snRNAseq

Kidneys from both WT mice and FibEGFR−/− mice at day 1 and day 3 after UUO were used with kidneys of 3–5 mice from same group being combined for analysis. Nuclei were isolated using Nuclei EZ lysis buffer. Nuclei (>20,000 nuclei per sample) were submitted for processing using the 10X Genomics platform. Libraries were prepared using P/N 1000121, 1000157, and 1000213 following the manufacturer's protocol. The libraries were sequenced using the NovaSeq 6000 with 150 bp paired end reads. RTA (version 2.4.11; Illumina) was used for base calling and analysis was completed using 10X Genomics Cell Ranger software v7.0.1. The Vanderbilt VANTAGE NGS sequencing core provided technical assistance for this work.

## RNAscope

RNA in situ hybridization (RNAscope, ACD Bio-Techne) was performed on FFPE sections using the RNAScope Multiplex Fluorescent Reagent v2 kit RED according to the manufacturer's instructions and our previous report[37]. Probes against mouse Acta2 (Mm-Acta2-C3, ACD 319531-C3) and mouse Areg (Mm-Areg-C1, ACD 430501-C1) were used.

## Confocal microscopy

Images were collected with confocal microscopy (Zeiss LSM980-Airyscan, objective 63×/1.0 NA Plan Apochromat oil).

**Single-nuclear RNA sequencing (snRNA seq) data processing.** Raw sequencing data were processed following the Chromium's Cell Ranger 3.1.0 pipeline with default parameters. The raw sequencing data (FASTQs files) were aligned to the mouse genome using STAR algorithm, and then gene-barcode matrices containing the barcoded cells and gene expression counts were generated and imported into the R package DropletUtils (version

1.16.0) to identification of cells from empty droplets, with removal of barcode-swapped pseudo- cells. Seurat (version 4.1.1) R toolkit was used for quality control and subsequent analysis. Unless specified, the default parameters were used for all functions. For quality control, genes that were

expressed in more than 10 cells, cells with detected genes between 500 to 7000, mitochondrial gene percentages less than 20%, and unique gene counts more than 1000 were kept. In the remaining cells, gene expression matrices were log normalized for each cell by the total expression and multiplied this by a scale factor (10,000 by default). Principal component analysis (PCA) for dimensional reduction was performed based on the highly variable genes (top 2000). Clusters were then visualized using the Uniform Manifold Approximation and Projection (UMAP) (ArXiv e-prints 1802.03426, 2018). Cell type identification was performed based on the expression of known cell markers[54–56].

**Gene set scoring analysis.** For gene set scoring analysis, fibroblast-related gene signatures and pathways based on the GSEA database were compared in Fib/MF cells subpopulations using the "AddModuleScore" function in the Seurat package[57]. The results were shown in the form of violin plots.

**GO analysisGene.** Ontology (GO) analysis was performed using the enrichGO function of the clusterProfiler (v3.16.1) package in R. Biological processes and 0.05 $P$ value cutoffs were chosen[58,59]. Heatmap indicated each GO term's significance (-log10 $P$ value).

**Cell-cell interaction/communication analysis**

Ligand Receptor Pair database CellChatDB was used for the cell–cell interaction/communication analysis through the standard pipeline of the R package CellChat version (1.4.0) to identify signaling patterns, predict pathways and information flow[60,61]. The results were shown in the forms of circle plots and bubble plots.Pseudotime analysis was performed using the Monocle package 2.26.0 with default parameters to reconstruct cell differentiation trajectories[60,62]. The obtained gene modules were analyzed and compared through GO analysis to show functional changes during cellular differentiation. The actual gestational time of each cell informed us of the start point of the pseudo-time in the first round of 'orderCells'. 'DDRTree' was applied to reduce dimensions and the visualization functions.

**Statistical analysis**

Statistical analyses were performed with GraphPad Prism 9 (GraphpadSoftware® Inc., La Jolla, CA, US). Data are presented as the mean ± S.E.M. Data were analyzed using 2 tailed Student's $t$ test or two-way ANOVA followed by Tukey's or Bonferroni's post hoc tests. A $P$ value less than 0.05 was considered significant. For each set of data, at least 4 different animals were examined for each condition. Collection, analysis, and interpretation of data were conducted by at least 2 independent investigators, who were blinded to the study.

**Study approval**

All animal experiments were performed in accordance with the guidelines and approval of the Institutional Animal Care and Use Committee of Vanderbilt University.

**Reporting summary**

Further information on research design is available in the Nature Portfolio Reporting Summary linked to this article.

## Data availability

All data required to reproduce the findings presented here can be found in the manuscript, figures and supplementary materials. Source data are provided with this paper. The raw data and processed gene expression data for 5 RNAseq samples described in this manuscript are available under accession number GSE244839. Source data are provided with this paper.

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

## Acknowledgements

These studies were supported by NIH grants, DK51265, DK95785, DK62794, P30DK114809 (R.C.H., M.Z.), VA Merit Award 00507969 (R.C.H.), The Vanderbilt Center for Kidney Disease, and Natural Science Foundation of China (No. 81870490) and Shanghai Pujiang Program (22PJ1409300) to Y.P. The Vanderbilt VANTAGE NGS sequencing core provided technical assistance for this work. VANTAGE is supported in part by CTSA Grant (5UL1 RR024975-03), the Vanderbilt Ingram Cancer Center (P30 CA68485), the Vanderbilt Vision Center (P30 EY08126), and NIH/NCRR (G20 RR030956).

## Author contributions

R.C.H. and M.Z. conceived the studies. S.C., Y.P., Y.W., J.T., X.F., S.W., S.K., M.J., W.L., X.D. and A.N. performed the studies. A.F. provided human samples, R.C.H., M.Z., Y.P. and S.C. analyzed the results and produced the figures. S.C., R.C.H., M.Z. and Y.P. wrote the manuscript and R.C.H., M.Z., A.T., J.P.A., M.H.W. and A.F. edited the manuscript.

## Competing interests

The authors declare no competing interests.
