## [Peer Review File · Nature Communications]

Epidermal Growth Factor Receptor Activation is Essential for Kidney Fibrosis DevelopmentREVIEWER COMMENTS

Reviewer #1 (Remarks to the Author):

The study by Cao et al. titled "Epidermal Growth Factor Receptor Activation is Essential for Kidney Fibrosis Development" provides valuable insights into the cell-type specific effects of EGFR signaling in various transgenic mouse models of kidney fibrosis. Through innovative approaches like iRhom2 KO mice and PDGFRB-specific Cre deletion in pericyte/fibroblasts, the authors demonstrate significant fibrosis amelioration in multiple mouse models. The study highlights the pivotal role of EGF signaling as an initiator event of fibrosis, with other pathways like TGF- β coming into play later. This early involvement of EGFR signaling may also be crucial in fibrosis development in other organs.

The authors utilized cutting-edge techniques such as transgenic mouse models, staining, single-cell RNA sequencing, and in-vitro assays to investigate the initiation of fibrosis and the pathways involved in different cell types. However, some aspects of the study warrant critical attention.

Major points:

1. Quantitative assessments of phosphorus-staining by immunofluorescence (Fig. 1 J) may introduce bias and errors. To improve the analysis, I recommend using western blot to assess the p-EGFR state in fibroblasts. Despite the challenges posed by low cell numbers, performing an immunoblot on FACS sorted cells would yield more convincing and less biased results.
2. The insights from single-cell RNA sequencing are somewhat limited and primarily confirmatory. It would be beneficial for the authors to explore the cell-cell communication between PTs and fibroblasts using PIC-Seq or Physical Interaction Sequencing in both WT and KO situations. This approach could provide more insights and even be performed at different time points to study the cell-cell communication trajectory.
3. While the study includes an impressive array of fibrosis models (IRI, UUO, folic acid,

adenine), it would be interesting to examine later time points of fibrosis. Investigating the effects of EGFR KO when fibrosis is already established to a certain degree could shed light on its potential ameliorative effects in later stages.

4. For PDGFR β FACS sorting, the authors used an APC-labeled antibody (Biolegend) that did not work efficiently in the reviewer's experiments. It is suggested that the authors utilize their PDGFR β labeled mouse line to label the cells with an endogenous fluorophore, which would significantly enhance the efficiency of the sorting process. Additionally, using BrdU or other recommended markers for cell proliferation analysis in vivo would be preferable over Ki-67.

5. From the antibody list it is unclear which ab the authors used for pEGFR in IF staining since for the Santa Cruz ab sc-12351 only WB is indicated while p-EGFR IF staining was performed in e.g. Fig 1 J.

6. A comparison of the relative strength of fibrosis reduction between the different models would be interesting based on the qPCR data.

7. While the description of the sn-RNA sequencing data in the additional methods section is sparse publishing the associated code and data is important and missing at the moment.

8. For trajectory analysis and cell-cell communication the authors use rather outdated methods. v2.0.0 is available for CellChatDB and should be updated. For

Minor:

- There appears to be a typo after citation 48. HB-EGF is known as "to63406" ...

- In the legend of Figure 8, there is a typo in the last sentence: "on the tday"

In conclusion, Cao et al.'s study provides important insights into EGFR's role in kidney fibrosis. By addressing the mentioned concerns and suggestions, the study's impact can be further enhanced, contributing significantly to the understanding of fibrosis initiation and

cell-type-specific responses to EGFR stimulation.

Reviewer #2 (Remarks to the Author):

Cao and Pan et al reported the role of EGFR in kidney fibrosis using a variety of approaches including histological and molecular assays on three genetic knockout models, single nucleus RNA-seq, and in vitro culture. The rich amount of data from different angles to demonstrate the role of EGFR make the whole study very convincing. In addition, as opposed to some other published snRNA-seq studies where the results are quite descriptive and diffuse, the interpretation of the snRNA-seq results converge to validate the changes in genes and pathways within fibroblasts from both wild-type (WT) and EGFR knockout mice. Overall, this study is quite comprehensive and presents novel findings in the field of kidney biology. Here are my comments to help improve the quality of the manuscript.

1. Did the authors examine the expression of the EGFR ligands? What cell types in the kidney, by secreting EGFR ligands, trigger the activation of fibroblast/pericyte during kidney fibrosis? It may help to find evidence from the single cell dataset and then use immunofluorescence to validate.

2. The authors did snRNA-seq only on the UUO model but the conclusion extends to other disease models such as IRI and folic acid models. As single cell RNA-seq datasets for these models are available, the authors should re-analyze those datasets to strengthen the main conclusion drawn from this study.

3. The analysis of human kidney data seems somewhat superficial. It would be beneficial for the authors to reexamine the fibroblast population from the published human CKD kidney scRNA-seq to evaluate the EGFR activation during fibrosis. This approach could elevate the impact of the current mouse-based findings.

4. Figure 4A: The sub-populations of fibroblasts in the UMAP appear to be intermixed, such as fib2 and MF2, despite the cell marker expression between fib2 and MF2 being quite distinct according to Figure 4B. It appears that the UMAP or clustering did not capture the

differences between the subtypes. The authors should consider adjusting the parameters when executing subclustering to determine if this inconsistency arises from inaccurate parameters used in clustering.

5. Fig4F: The authors should normalize the fibroblast number by the total number of cells included in the WT and KO datasets. A stacked bar plot following normalization to visualize the percentage of cells in each subtype between WT and KO might provide a more accurate representation than the existing scatter plot.

Reviewer #3 (Remarks to the Author):

In this study, Cao et al studied the mechanism of EGFR-mediated renal fibrosis and found that EGFR expression increased in interstitial fibroblasts/myofibroblasts in fibrotic kidneys. Deletion of *Rhbdf2* (*iRhom2*), a member of the Rhomboid family that regulates ADAM17-mediated release of membrane-anchored proteins, including EGFR ligands, inhibited interstitial fibrosis. Further, EGFR is necessary for the initial pericyte/fibroblast migration and proliferation prior to subsequent myofibroblast transformation by TGF- β or other profibrotic factors. Although the study provides some new insights into the biological functions of EGFR in regulating renal fibrosis, there are several concerns on this study.

1, Figure 1D shows that there are many α -SMA (+) fibroblasts, only few of them express *Rhbdf2* while deletion of *Rhbdf2* reduces a half level of profibrotic molecules (Figure 1E, F, G). On this basis, it is hard to imagine that deletion of *Rhbdf2* from renal fibroblasts contributes to such a larger level of reduction of renal fibrotic proteins.

2, IF staining in Figure 1J is poor, it seems that p-EGFR is most expressed in the cytosol rather than in the cellular membrane. It is better to examine the EGFR expression using confocal microscopy.

3. Figure 3 legend indicates the activation of EGFR signaling in myofibroblasts following UOU injury, but only co-staining of EGFR and α -SMA and mRNA levels of EGFR legends are shown, lack of analysis of EGFR phosphorylation.

4. PCNA, as a marker of cell proliferation, is not only expressed in the fibroblasts, but should also be expressed in renal epithelial cells. Figure 9 shows that few of myofibroblasts

expresses PCNA, but there are no renal epithelial cells expressing PCNA, in WT animals with UUO. This is because that PCNA(+) renal tubular cells was not seen in such a small area of the kidney section? It is likely that PCNA(+) renal tubular cells be seen in the early of UUO injured kidneys, especially the kidney sample from 3 days are used.

5. EGFR is usually expressed in renal epithelial cells, but this study shows that its upregulated in renal interstitial fibroblasts/pericytes, it would be interesting to know what mechanisms responsible for increased EGFR in renal interstitial fibroblasts/pericytes following fibrotic injury.

REVIEWER COMMENTS

Reviewer #1 (Remarks to the Author):

The study by Cao et al. titled "Epidermal Growth Factor Receptor Activation is Essential for Kidney Fibrosis Development" provides valuable insights into the cell-type specific effects of EGFR signaling in various transgenic mouse models of kidney fibrosis. Through innovative approaches like iRhom2 KO mice and PDGFRB-specific Cre deletion in pericyte/fibroblasts, the authors demonstrate significant fibrosis amelioration in multiple mouse models. The study highlights the pivotal role of EGF signaling as an initiator event of fibrosis, with other pathways like TGF- β coming into play later. This early involvement of EGFR signaling may also be crucial in fibrosis development in other organs.

The authors utilized cutting-edge techniques such as transgenic mouse models, staining, single-cell RNA sequencing, and in-vitro assays to investigate the initiation of fibrosis and the pathways involved in different cell types. However, some aspects of the study warrant critical attention.

Major points:

1. Quantitative assessments of phosphorus-staining by immunofluorescence (Fig. 1 J) may introduce bias and errors. To improve the analysis, I recommend using western blot to assess the p-EGFR state in fibroblasts. Despite the challenges posed by low cell numbers, performing an immunoblot on FACS sorted cells would yield more convincing and less biased results.

Answers: *We are grateful for the reviewer's suggestion. We are concerned that the FACS procedure will lead to dephosphorylation of p-EGFR. Therefore, we employed confocal microscopy to evaluate p-EGFR in WT and iRhom2^{-/-} mice 7 days after UUO. As indicated in new **Figure 1J**, p-EGFR is clearly present in basolateral membranes of epithelial cells (arrows) as well as in α -SMA⁺ fibroblasts (asterisk) in the WT mice. In iRhom2^{-/-} mice, although p-EGFR is also clearly present in basolateral membranes of epithelial cells, its expression in α -SMA⁺ fibroblasts is minimal.*

2. The insights from single-cell RNA sequencing are somewhat limited and primarily confirmatory. It would be beneficial for the authors to explore the cell-cell communication between PTs and fibroblasts using PIC-Seq or Physical Interaction Sequencing in both WT and KO situations. This approach could provide more insights and even be performed at different time points to study the cell-cell communication trajectory.

Answers: *We are thankful for this wonderful suggestion, and we will be considering this suggestion in future studies.*

3. While the study includes an impressive array of fibrosis models (IRI, UUO, folic acid, adenine), it would be interesting to examine later time points of fibrosis. Investigating the effects of EGFR KO when fibrosis is already established to a certain degree could shed light on its potential ameliorative effects in later stages.

Answers: *We are thankful for the reviewer's suggestions. In the revised manuscripts, we have performed immunoblotting of α -SMA, collagen I, and Fibronectin after treatment with IRI for 28 days (**Figure 6E**) and with folic acid for 19 days (**Figure 7I**), as well as quantitative immunostaining for α -SMA after 7-day UUO (**Figure 5E**), 28-day IRI (**Figure 6F**), 19-day folic acid (**Figure 7J**), and 14-day adenine (**Suppl Figure S8G**).*

4. For PDGFRb FACS sorting, the authors used an APC-labeled antibody (Biolegend) that did not work efficiently in the reviewer's experiments. It is suggested that the authors utilize their PDGFR β labeled mouse line to label the cells with an endogenous fluorophore, which would significantly enhance the efficiency of the sorting process. Additionally, using BrdU or other recommended markers for cell proliferation analysis in vivo would be preferable over Ki-67.

Answers: We appreciate the reviewer's suggestion. We re-analyzed our data using endogenous mCherry as marker of myofibroblasts to analyze myofibroblasts and Edu incorporation to determine mCherry+ cell proliferation. As indicated in new **Figure 9 B and C**, both mCherry+ myofibroblasts and Edu+ mCherry+ (proliferating) myofibroblasts were significantly lower in FibEGFR-/- mice compared to WT mice 3 days after UUO. Of note, myofibroblast number evaluated using the endogenous fluorophore is almost 10-fold higher and the variation is smaller than our original data. In addition, Edu+ mCherry+ cell number was also doubled using the endogenous fluorophore.

5. From the antibody list it is unclear which ab the authors used for pEGFR in IF staining since for the Santa Cruz ab sc-12351 only WB is indicated while p-EGFR IF staining was performed in e.g. Fig 1 J.

Answers: We are sorry for this confusion. We used the recombinant anti-EGFR (phosphor Y1068, EP774Y, ab40815) for pEGFR IF. We now add related information in the revised manuscript.

6. A comparison of the relative strength of fibrosis reduction between the different models would be interesting based on the qPCR data.

Answers: The correlation between EGFR deletion and Fibrosis reduction is provided in the revised manuscript (**Suppl Figure S14**).

7. While the description of the sn-RNA sequencing data in the additional methods section is sparse publishing the associated code and data is important and missing at the moment.

Answers:

The raw and processed gene expression data in this paper have been deposited into the NCBI GEO database:GEO.

No new algorithms were developed for this manuscript. CellChat code is available in <https://github.com/sqjin/CellChat>. Monocle can download from <http://bioconductor.org/packages/release/bioc/html/monocle.html>.

8. For trajectory analysis and cell-cell communication the authors use rather outdated methods. v2.0.0 is available for CellChatDB and should be updated.

Answers: Thank you for this information. In the revised manuscript, we state that "The R package CellChat is used, and CellChatDB is the built-in database in CellChat without a separate version" in the Figure legend.

For Minor:

- There appears to be a typo after citation 48. HB-EGF is known as "to63406" ...

Answers: We apologize for the typo and have deleted 63406 in the revised manuscript.

- In the legend of Figure 8, there is a typo in the last sentence: "on the tday".

Answers: We apologize for the typo and have corrected it in the revised manuscript.

In conclusion, Cao et al.'s study provides important insights into EGFR's role in kidney fibrosis. By addressing the mentioned concerns and suggestions, the study's impact can be further enhanced, contributing significantly to the

understanding of fibrosis initiation and cell-type-specific responses to EGFR stimulation.

Reviewer #2 (Remarks to the Author):

Cao and Pan et al reported the role of EGFR in kidney fibrosis using a variety of approaches including histological and molecular assays on three genetic knockout models, single nucleus RNA-seq, and in vitro culture. The rich amount of data from different angles to demonstrate the role of EGFR make the whole study very convincing. In addition, as opposed to some other published snRNA-seq studies where the results are quite descriptive and diffuse, the interpretation of the snRNA-seq results converge to validate the changes in genes and pathways within fibroblasts from both wild-type (WT) and EGFR knockout mice. Overall, this study is quite comprehensive and presents novel findings in the field of kidney biology. Here are my comments to help improve the quality of the manuscript.

1. Did the authors examine the expression of the EGFR ligands? What cell types in the kidney, by secreting EGFR ligands, trigger the activation of fibroblast/pericyte during kidney fibrosis? It may help to find evidence from the single cell dataset and then use immunofluorescence to validate.

Answers: After UUO, total kidney mRNA levels of EGFR ligands including HB-EGF, AREG as well as ADAM17 increased (Figure 2E, F, J). We performed IF for HB-EGF and ADAM17 and RNAscope for AREG (Suppl Figure S4).

- HB-EGFR is expressed at low levels in epithelial cells under normal condition. After UUO, HB-EGF expression increased in both epithelial cells and myofibroblasts (colocalization with α -SMA).
- We did not find a good antibody for AREG that worked for IF. We performed RNAscope for AREG mRNA expression. AREG mRNA was expressed in epithelial cells (red arrows) and in myofibroblasts (white arrows, colocalization with Acta2) at day 7 after UUO in WT mice.
- Of note, 3 days after UUO, ADAM17 was primarily expressed in myofibroblasts, and its expression decreased in FibEGFR^{-/-} mice.

2. The authors did snRNA-seq only on the UUO model but the conclusion extends to other disease models such as IRI and folic acid models. As single cell RNA-seq datasets for these models are available, the authors should re-analyze those datasets to strengthen the main conclusion drawn from this study.

Answers: We re-analyzed scRNAseq data published by Humphrey's group in PNAS in 2020 and it clearly indicated that EGFR and its ligands Hb-egf, Areg, and well as Adam17 increase immediately after IRI (Suppl Figure S12). #17550}.

3. The analysis of human kidney data seems somewhat superficial. It would be beneficial for the authors to reexamine the fibroblast population from the published human CKD kidney scRNA-seq to evaluate the EGFR activation during fibrosis. This approach could elevate the impact of the current mouse-based findings.

Answers: We re-analyzed the expression of EGFR in fibroblast populations of human CKD kidney scRNA-seq published by Humphreys' group. It clearly demonstrated that human fibroblasts and myofibroblasts express high EGFR mRNA levels (Suppl Figure S13).

4. Figure 4A: The sub-populations of fibroblasts in the UMAP appear to be intermixed, such as fib2 and MF2, despite the cell marker expression between fib2 and MF2 being quite distinct according to Figure 4B. It appears that the UMAP or clustering did not capture the differences between the subtypes. The authors should consider adjusting the parameters when executing subclustering to determine if this inconsistency arises from inaccurate parameters used in clustering.

Answers: Because this is a clustering of the same type of cells, the expression gap between cells is relatively small. In revised manuscript, we use the tSNE plot, which can separate sub-populations of fibroblasts quite well (Figure 4A).

5. Fig4F: The authors should normalize the fibroblast number by the total number of cells included in the WT and KO datasets. A stacked bar plot following normalization to visualize the percentage of cells in each subtype between WT and KO might provide a more accurate representation than the existing scatter plot.

Answers: *As suggested by the reviewer, we now present fibroblast number with a stacked bar plot (Figure 4F).*

Reviewer #3 (Remarks to the Author):

In this study, Cao et al studied the mechanism of EGFR-mediated renal fibrosis and found that EGFR expression increased in interstitial fibroblasts/myofibroblasts in fibrotic kidneys. Deletion of *Rhbdf2* (*iRhom2*), a member of the Rhomboid family that regulates ADAM17-mediated release of membrane-anchored proteins, including EGFR ligands, inhibited interstitial fibrosis. Further, EGFR is necessary for the initial pericyte/fibroblast migration and proliferation prior to subsequent myofibroblast transformation by TGF- β or other profibrotic factors. Although the study provides some new insights into the biological functions of EGFR in regulating renal fibrosis, there are several concerns on this study.

1, Figure 1D shows that there are many α -SMA (+) fibroblasts, only few of them express *Rhbdf2* while deletion of *Rhbdf2* reduces a half level of profibrotic molecules (Figure 1E, F, G). On this basis, it is hard to imagine that deletion of *Rhbdf2* from renal fibroblasts contributes to such a larger level of reduction of renal fibrotic proteins.

Answers: *Thank you for this point. In this study, we used global *iRhom2*^{-/-} mice. Therefore, *iRhom2* from fibroblasts as well as other cell types may all contribute to the phenotype.*

2, IF staining in Figure 1J is poor, it seems that p-EGFR is most expressed in the cytosol rather than in the cellular membrane. It is better to examine the EGFR expression using confocal microscopy.

Answers: *At the reviewer's suggestion, we performed p-EGFR IF using confocal microscopy and clearly show p-EGFR is expressed in plasma membrane (Figure 1J)*

3. Figure 3 legend indicates the activation of EGFR signaling in myofibroblasts following UUO injury, but only co-staining of EGFR and α -SMA and mRNA levels of EGFR legends are shown, lack of analysis of EGFR phosphorylation.

Answers: *In the revised manuscript, we included p-EGFR expression in WT and *FibEGFR*^{-/-} mice at day 0, day 1 and day 3 after UUO. p-EGFR was expressed primarily in epithelial cells at day 0 and day 1 after UUO in both WT and *FibEGFR*^{-/-} mice. Three days after UUO, p-EGFR expression in myofibroblasts significantly increased in WT mice, but was minimal in myofibroblasts in *FibEGFR*^{-/-} mice (Suppl Figure S6).*

4. PCNA, as a marker of cell proliferation, is not only expressed in the fibroblasts, but should also be expressed in renal epithelial cells. Figure 9 shows that few of myofibroblasts express PCNA, but there are no renal epithelial cells expressing PCNA, in WT animals with UUO. This is because that PCNA(+) renal tubular cells were not seen in such a small area of the kidney section? It is likely that PCNA(+) renal tubular cells be seen in the early of UUO injured kidneys, especially the kidney sample from 3 days are used.

Answers: *Thank you for the reviewer's comments. We restained kidney with PCNA from 3 days after UUO. As indicated in new Figure 9D, PCNA was clearly expressed in both epithelial cells (white arrow) and myofibroblasts (red arrows). PCNA+ myofibroblasts were significantly decreased in *FibEGFR*^{-/-} mice 3 days after UUO.*

5. EGFR is usually expressed in renal epithelial cells, but this study shows that it is upregulated in renal interstitial fibroblasts/pericytes, it would be interesting to know what mechanisms are responsible for increased EGFR in renal interstitial fibroblasts/pericytes following fibrotic injury.

Answers: *Both EGFR ligands and ADAM17 were increased in the kidneys, as indicated by total kidney qPCR. In addition, HB-EGF expression in myofibroblasts increased after UUO (Suppl Figure S4). ADAM17 was found to be expressed in myofibroblasts in the kidney and its expression was lower in *FibEGFR*^{-/-} than WT mice after UUO (Suppl*

Figure S4). Therefore, increased EGFR ligands HB-EGF and AREG as well ADAM17-mediated release of EGFR ligands may all contribute to increased EGFR activation in myofibroblasts.

REVIEWERS' COMMENTS

Reviewer #1 (Remarks to the Author):

All comments and questions have been addressed adequately by the authors.

Reviewer #2 (Remarks to the Author):

My concerns have been addressed. The manuscript have been greatly improved in this revision. I don't have further comments.

Reviewer #3 (Remarks to the Author):

All my concerns have been addressed except this one " EGFR is usually expressed in renal epithelial cells, but this study shows that it is upregulated in renal interstitial fibroblasts/pericytes, it would be interesting to know what mechanisms responsible for increased EGFR in renal interstitial fibroblasts/pericytes following fibrotic injury."

The author's answer was about EGFR activation rather than changed EGFR levels.

We appreciate the careful review and the positive comments of all reviewers. Reviewer #3 had one additional concern:

All my concerns have been addressed except this one " EGFR is usually expressed in renal epithelial cells, but this study shows that it is upregulated in renal interstitial fibroblasts/pericytes, it would be interesting to know what mechanisms responsible for increased EGFR in renal interstitial fibroblasts/pericytes following fibrotic injury."

In response, we have provided the following sentence in the discussion, with an appropriate reference :

In addition, we found that there was increased EGFR expression in kidney interstitial fibroblasts from both humans and experimental animals with tubulointerstitial fibrosis. EGFR activation has previously been shown to induce a feed-forward response to further increase EGFR expression¹⁴.